



# Measurement report: In-depth characterization of ship emissions during operations in a Mediterranean port

Lise Le Berre [1,2,3], Brice Temime-Roussel [1], Grazia Maria Lanzafame [1], Barbara D'Anna [1], Nicolas Marchand [1], Stéphane Sauvage [4,5], Marvin Dufresne [4], Liselotte Tinel [4], Thierry Leonardis [4], Joel Ferreira de Brito [4], Alexandre Armengaud [3], Grégory Gille [3], Ludovic Lanzi [3], Romain Bourjot [3] and Henri Wortham [1]

[1] Aix Marseille Univ., CNRS, LCE, Marseille, France
[2] ADEME, French Agency for ecological transition, Angers, France
[3] AtmoSud, Regional Network for Air Quality Monitoring of Provence-Alpes-Côte-d'Azur, Marseille, France
[4] CERI EE, Centre for Education, Research and Innovation in Energy and Environment, IMT Nord Europe, Institut Mines-Télécom, Univ. Lille, Lille, France
[5] Univ. Grenoble-Alpes, CNRS, IRD, INP, INRAE, IGE, Grenoble, France

*Correspondence to*: Lise Le Berre (lise.le-berre@univ-amu.fr) and Brice Temime-Roussel (brice.temime-roussel@univ-amu.fr)

**Abstract.**

A summertime field campaign was conducted in Marseille, one of the major cruise and ferry ports in the Mediterranean to provide comprehensive analysis of in-port ship emissions. High-temporal-resolution data were simultaneously collected from two monitoring stations deployed in the port area to examine the composition in both the gas and the particulate phases. More than 350 individual plumes were captured from a variety of ships and operational phases. Gaseous emissions are predominantly composed of $NO_X$ (86%) and CO (12%), with $SO_2$ and $CH_4$ each accounting for about 1%. Although NMVOCs make up less than 0.1% of the gaseous phase, they can be as high as 10% under specific operational conditions. Submicron particles ($PM_1$) are mainly composed of organics (75%), black carbon (21%), and sulphate (4%) not balanced with ammonium. Among the ship-related characteristics investigated, the operational phase is the most influential with a threefold increase on $PM_1$ emissions, along with higher relative contributions of BC and sulphate, and detection of vanadium, nickel, and iron during "manoeuvring/navigation" compared to "at berth". Pollutant levels in the port are higher than those found at the urban background site, with average concentrations of NOx, $PM_1$, and particle number (PN) up to twice as high. Analysis of the maximum concentrations reveals that pollutants such as $SO_2$ and trace metals, including Vanadium and Nickel, are 2 to 10 times higher in the port area. This study provides robust support for enhancing source apportionment and emission inventories both of which are crucial for assessing air, health and climate impacts of shipping.

Keywords: Ships, Emission factors, Port, Gases, $PM_1$, trace metal, Air Quality



## 1. Introduction

Maritime transport is one of the most economical modes of transport in terms of tonnes of goods or passengers carried. It has grown significantly in recent years (Toscano and Murena, 2019) due to the increase in international manufacturing, trade and tourism (Sorte et al., 2020). Projections forecast sustained growth, with freight transport doubling in 2030 compared to 2020 (UNCTAD, 2023). While this mode of transport is a key contributor to social and economic development worldwide (Bagoulla and Guillotreau, 2020; Eyring et al., 2010), it also negatively impacts global climate and air quality in ports and coastal areas

(Aardenne et al., 2013; Toscano, 2023; Viana et al., 2020). Moreover, several studies demonstrated that emissions from maritime transport have negative effects on human health (Corbett et al., 2007; Kiihamäki et al., 2024; Liu et al., 2016; Mueller et al., 2023; Oeder et al., 2015; Wu et al., 2020; Zhang et al., 2021).

Given that other sectors (such as industry, road traffic, and heating) have significantly reduced their emissions, maritime transport now accounts for a growing proportion of total emissions. In 2018, shipping was responsible for approximately one

billion tons of carbon dioxide ($CO_2$), a potent greenhouse gas, accounting for about 3% of global anthropogenic emissions (IMO, 2020). Projections suggest that by 2050, maritime transport could represent as much as 15% of global $CO_2$ emissions (Serra and Fancello, 2020). Shipping is also a relevant source of atmospheric pollutants including nitrogen oxides ($NO_X$), sulphur oxides ($SO_X$), carbon monoxide (CO), volatile organic compounds (VOC) and particulate matter (PM) (Johansson et al., 2017; Sorte et al., 2020; Toscano, 2023). It is estimated that shipping accounts for 20-28% of both global and European

$NO_X$ and $SO_X$ emissions, and for 5% of $PM_{10}$ (PM smaller than 10 µm) of the European emissions (Contini and Merico, 2021; Russo et al., 2018). Research on particles emitted by ships indicates that they are mostly submicron, with diameters under 100 nm (Alanen et al., 2020; Jeong et al., 2023; Kuittinen et al., 2021), and are composed of black carbon (BC), organic aerosols (OA) including polycyclic aromatic hydrocarbons (PAH), sulphate ($SO_4^{2-}$), as well as, to a lesser extent metals. Although PAH and metals are emitted in smaller quantities, they are recognized for their strong health impacts (Briffa et al., 2020; Fridell et

al., 2008). Ship emissions are also important sources of gaseous precursors leading to the formation of secondary organic and inorganic aerosol at local and regional scales (Celik et al., 2020; Karl et al., 2023; Lanzafame et al., 2022; Liu et al., 2022; Pérez et al., 2016).

Numerous legislative efforts have therefore been made, at both global and local levels, to reduce emissions of atmospheric pollutants linked to maritime transport. The International Maritime Organisation (IMO) adopted the International Convention

for the Prevention of Pollution from Ships (MARPOL Convention), Annex VI of which deals with the mitigation of air pollution. This convention limits emissions of $SO_X$ and NOx. Since the 1st of January 2020, the sulphur content in fuels should not exceed 0.5% m/m (compared with 3.5% m/m for the previous limit). Since 2015, it must even be less than 0.1% m/m in emission control areas (ECA) (IMO, 2021). Since 1 January 2021, nitrogen oxide emissions are also controlled in ECA for ships built after 2021 (IMO, 2021). From 1 January 2025 onwards, ECA-MED will make compulsory to use fuels with sulphur

content of less than 0.1% m/m in the Mediterranean (UNEP/MAP, 2021). The European Union (EU) also restricts the sulphur





content in fuel used by ships to 0.1% while they are docked or anchored in all EU ports, with an exception for ships staying no longer than two hours (EU, 2016).

These regulations have led to significant progresses in ship engines and to the introduction of after-treatment devices, such as Selective Non-Catalytic Reduction (SCR) systems and scrubbers to reduce emissions of NOx and $SO_2$ respectively. These

systems enable some ships to continue using fuels with sulphur content exceeding 0.5% or 0.1% in ECA. These developments result in a wide array of possible combinations of after-treatment devices, fuels and engines used. The after-treatment devices limit the quantity of pollutants emitted but also change their chemical composition (Fridell and Salo, 2016; Jeong et al., 2023; Kuittinen et al., 2024; McCaffery et al., 2021; Timonen et al., 2017, 2022; Winnes et al., 2020). In addition, various studies have demonstrated the impact of switching to cleaner fuels on shipping emissions (Alanen et al., 2020; Gysel et al., 2017;

Jeong et al., 2023; Kuittinen et al., 2024; Lehtoranta et al., 2019; McCaffery et al., 2021; Yang et al., 2022; Zetterdahl et al., 2016). For example, Zetterdahl et al. (2016) showed that switching from heavy fuel oil (HFO) with a sulphur content (FSC) of 0.5% to marine diesel oil (MDO) with 0.1% sulphur on a specific ship resulted in a 67% reduction in total particulate mass, but no reduction in the number of particles. Kuittinen et al. (2024) detailed the changes in particulate chemical composition, including PAH and metals, for a cruise ship using HFO containing 0.7% sulphur and MGO containing 0.1% sulphur. Finally,

others studies have emphasized the benefits of different engine categories and upgrades on reducing ship emissions (Fridell et al., 2008; Grigoriadis et al., 2021b; McCaffery et al., 2021; Sugrue et al., 2022; Xiao et al., 2018). Sugrue et al. (2022) observed that newer engines (built after 2016) emit three times less BC than engines built before 2000. Grigoriadis et al. (2021b) showed in their emission factors review that, among engines built before 2016, slow-speed diesel (SSD) engines emit 1.5 and 2 times more $NO_X$ than medium-speed diesel (MSD) and high-speed diesel (HSD) engines, respectively. They also underscored the

lack of data regarding emissions from auxiliary engines, which are used by ships while docked. McCaffery et al. (2021) nonetheless showed that NOx emissions from the main engine (SSD) of a container ship running on MGO were twice as high as those from its auxiliary engines (MSD) also operating on MGO. However, this difference cannot be attributed to the engine being main or auxiliary, but rather to the engine category (SSD/MSD), as highlighted by Grigoriadis et al. (2021b).

Based on the aforementioned studies, it can be concluded that research often focuses on specific ships or on a limited set of

pollutants (typically $NO_X$, $SO_2$, PM), and have usually been conducted during open-sea operations, thus overlooking the specific characteristics of port operations. Studying emissions during port operations is crucial, as these emissions significantly differ and more directly impact air quality and public health in port cities (Toscano, 2023; Viana et al., 2020). In fact, the contribution of ship emissions becomes more prominent as the area of focus narrows—from less than 5% for $PM_{2.5}$ on a global scale (Crippa et al., 2019), to 15% in the Mediterranean region (Fink et al., 2023), and up to 60% within port areas, such as

the French port of Calais (Ledoux et al., 2018), where emissions from port operations become more significant. While docked, ships primarily use their auxiliary engines at optimal and stable loads, resulting in relatively steady emissions. In contrast, during manoeuvring or navigation, ships mostly rely on their main engines, which often operate at low and unstable loads, leading to fluctuating emissions. Engine startups can also cause significant emission spikes. Additionally, as previously mentioned, the use of exhaust treatment systems and different fuel types further complicates the analysis.




Considering all these elements, this study provides the physical and chemical characteristics of plumes emitted by various ships during port operations in Marseille, one of the largest ports in the Mediterranean. This coastal city faces significant anthropogenic pressure, which contributes to concerning levels of atmospheric pollution, especially fine particles (Chazeau et al., 2021). The analysis is based on high temporal resolution measurements to characterize the overall composition of ship

emissions, both gaseous and particulate, with a particular focus on submicronic particle composition. To achieve this, ship emission plumes were identified by cross-referencing historical Automatic Identification System (AIS) data on ship locations with meteorological conditions. Emission factors (EF) have then been calculated accounting for ship category, operating phase and plume age, providing crucial data for assessing the impact of ship emissions in coastal areas.

## 2. Measurements and methods

### 2.1. Measurement sites

The measurement campaign took place in the summer of 2021 (from 30 May to 3 July) in the port of Marseille (Grand Port Maritime de Marseille, GPMM) on the French Mediterranean coast. This port is located alongside the central area of Marseille, the second-largest city in France, in terms of population (INSEE, 2020). It is one of the major passenger ports in the Mediterranean, with yearly flux of 3 million of passengers via 500 cruise ships and 2200 ferries (or also termed ro-ro passenger

ships) calls, respectively (Marseille Fos Port, 2022, 2023). The port also receives nearly 1000 cargo ship calls, handling 77 million tonnes of goods (Marseille Fos Port, 2023). Freight ship traffic remains relatively constant throughout the year, with an average of 80 calls per month, while cruise and ferry traffic intensifies between April and October, with an average of 55 and 200 calls per month respectively, compared to 25 and 150 for the rest of the year.

In 2020, worldwide maritime ship traffic, and consequently that of Marseille, suffered a sharp decline due to restrictions linked

to the Covid-19 pandemic, especially for cruise ships (-92% in Marseille (Marseille Fos Port, 2022)) which remained docked for just over a year. However, in 2021, a strong recovery in traffic was recorded, with the exception of cruise traffic, which only picked up in July, and the 2019 levels were reached again in 2022 (Marseille Fos Port, 2022, 2023).

Measurements were conducted simultaneously at two stations located inside the port area, along the berths (Figure 1). The two stations were chosen based on the analysis of historical weather data and exploratory measurements of air quality inside the

port, according to the following criteria: i) to be as close as possible to ship emissions and therefore to shipping lanes (Figure 1) ii) to limit the influence of other sources iii) to maximise the probability of the station being downwind of plumes and iv) to capture plumes representative of the diversity of ships accessing to the port of Marseille from the north or south. Cruises and cargo ships access the port from the North while ferries access from both the North and South. The north channel is mainly used by ferries to and from Corsica, while the south channel is used by ferries to and from international destinations in addition

to Corsica.



The first station, labelled PEB, was located on a seawall, 150 m from the northern access seaway to the port, less than 700 m south-east of the cruise terminals, 1 200 m south of the container terminal and about 800 m north-west of the ferry berths to Corsica (43°20'6.89" N ; 5°20'21.76" E ; 5 msl).

The second station, labelled MAJOR, was located along the "Joliette" berth, the main access road to the port, 250 m from the access lane to the port via the southern pass, less than 200 m north-east of the luxury cruise terminal, 150 m east of the ferry berths to Corsica and around 350 m south-east of the berthing quays for ships travelling to and from North Africa (43°18'0.51" N ; 5°21'48.01" E ; 5 msl).

These two stations are located 2,500 m west and 5,500 m northwest of Marseille's urban background pollution reference station (MRS-LCP) (Figure 1).

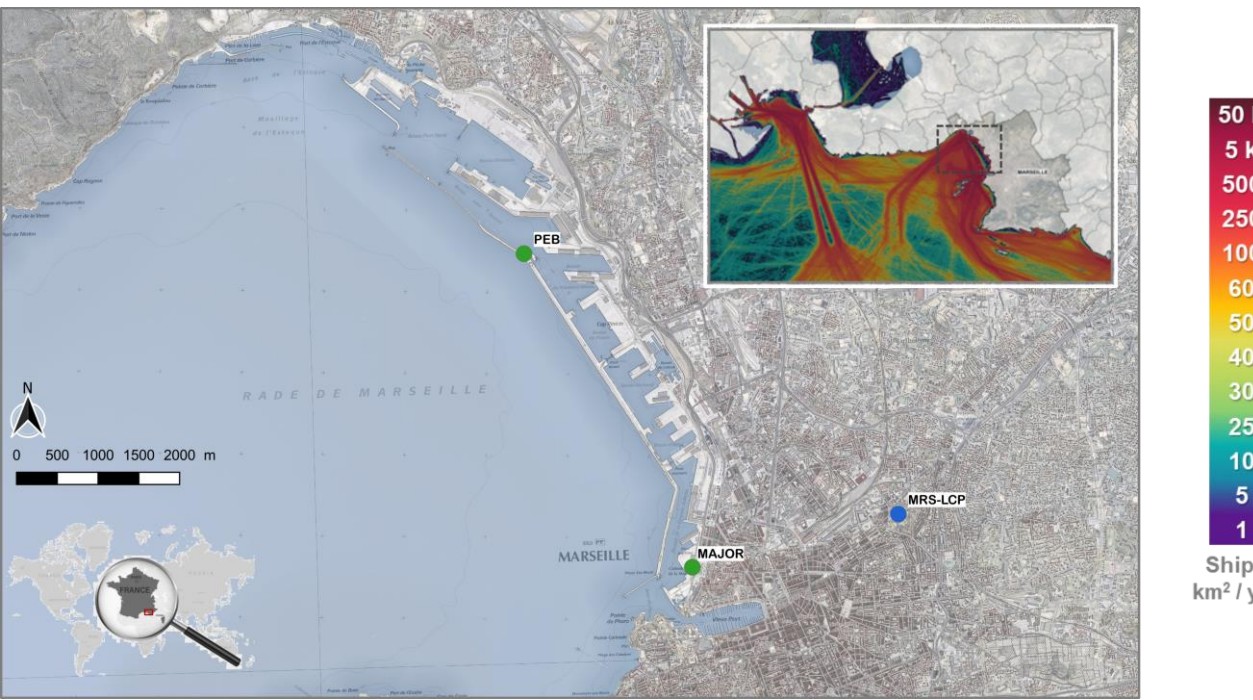

**Figure 1. (a) Map of the port of Marseille with the measurement stations (green filled circle)—PEB near the northern access lane and MAJOR near the southern access lane— and the urban background reference pollution station MRS-LCP (blue filled circle). The inset map shows ship traffic density on a larger scale; with the colour bar indicating the number of ships passing per year per square kilometre (MarineTraffic, 2022). Maps taken from Google satellite images (© Google Maps) and topographic map SCAN 25 (©IGN – 2022).**

## 2.2. Instrumentation

Table 1 lists the on-line instruments that were housed in the two measurement stations. The technical specifications (flow rate, detection limits and uncertainties) and the quality controls (calibration and instrumental background) carried out to ensure the accuracy of the measurements are presented in Table S1 and Table S2. Individual sampling lines were used for most instruments with an air intake at about 4 meters above the ground level.





**Table 1.  Overview of instruments deployed during the field campaign**

| | Measured Quantity | Instrument | Size Range | Temporal resolution | PEB [1] | MAJOR [1] |
|---|---|---|---|---|---|---|
| **PARTICULATE PHASE** | **Particle number (PN)** Particle number concentration | CPC TSI 3776 (TSI) | 2.5 nm – 3 µm [2] | 1 s | X | |
| | | Envi CPC 200 (PALAS) [6] | 7 nm – 2.5 µm [2] | 1 s | | X |
| | **Particle size distribution** Particle number concentration | SMPS 3936 (CPC 3775 - Classifier 3080 - Long DMA) (TSI) | 15 nm – 660 nm [3] | 2 min | X | |
| | | SMPS 3936 (CPC 3776 - Classifier 3080 - Long DMA) (TSI) | 15 nm – 660 nm [3] | 2 min | | X |
| | **Particle size distribution** Particle number and mass concentration (PN, $PM_1$, $PM_{2.5}$, $PM_{10}$) | OPC model 1.109 (Grimm Aerosol Technik) | 0.25 µm – 32 µm [4] | 1 min | X | X |
| | **Black Carbon (BC)** Particle mass concentration | MAAP 5012 (ThermoFisher) [5] | <1 µm [2] | 1 min | X | |
| | | AE33 (Aerosol Magee Scientific) [5] | <1 µm [2] | 1 min | | X |
| | **Non refractory chemical composition** Particle mass concentration | HR-ToF-AMS, (Aerodyne Research) [5] | 30 nm – 600 nm [2] | 30 s | X | |
| | **Metals composition** Particle mass concentration | Xact 625i (Cooper Environmental) [5] | <1 µm [2] | 30 min | X | |
| **GAS PHASE** | **Non-Methane Volatile Organic Compounds (NMVOC)** Gazeous concentration | PTR-ToF-MS 8000 (Ionicon Analytik) [7] | n/a | 10 s | X | |
| | **Sulphur dioxide ($SO_2$)** Gazeous concentration | AF22 (Environnement SA) [8] | n/a | 10 s | X | |
| | | 100E (Teledyne API) [8] | n/a | 10 s | | X |
| | **Nitrogen oxides ($NO_X$, NO, $NO_2$)** Gazeous concentration | 200E (Teledyne API) [8] | n/a | 10 s | X | X |
| | **Ozone ($O_3$)** Gazeous concentration | 400E (Teledyne API) [8] | n/a | 10 s | X | X |
| | **$CO_2$, CO, $CH_4$** Gazeous concentration | G2401 (PICARRO,) [9] | n/a | 5 s | X | X |
| | **Ammoniac ($NH_3$)** Gazeous concentration | G2103 (PICARRO) [8] | n/a | 5 s | X | |
| **AUXILIARY DATA** | **Wind speed (ws), wind direction (wd), Temperature (T)** Meteorological data | Weather station (2D) | n/a | 1 min | X | |
| | | Weather station (3D sonic) | n/a | 10 s | | X |

(1) Columns indicate at which station the instruments were operated; (2) aerodynamic diameter; (3) electrical mobility diameter; (4) optical diameter; (5) equipped with a PM1 cut-off inlet; (6) equipped with a PM2.5 cut-off inlet; (7) equipped with 1/16" silcosteel tubing; (8) equipped with ¼" PTFE tubing; (9) equipped with ¼" Synflex tubing





### 2.2.1. Particle phase measurements

The chemical composition of the submicronic fraction of the aerosol was studied using 3 different analysers. Concentrations of non-refractory species, namely sulphate ($SO_4^{2-}$), nitrate ($NO_3^-$), ammonium ($NH_4^+$), chloride ($Cl^-$) and organic matter (OA), were measured at the PEB station, with a high-resolution time-of-flight aerosol mass spectrometer (HR-ToF-AMS, Aerodyne) with a 30 s time resolution (DeWitt et al., 2015). The metal composition of the submicronic particles was also determined on the PEB station using an online Energy-Dispersive X-ray Fluorescence (EDXRF) spectrometer (Xact 625i, Cooper

Environment) (Tremper et al., 2018) with a 30 min time resolution. Equivalent Black Carbon (BC) measurements at 1 min time resolution were performed with a multiangle absorption photometer (MAAP 5012, ThermoFischer) (Petzold and Schönlinner, 2004) at the PEB station and with a dual-spot seven-wavelength aethalometer (AE33, Magee Scientific) (Drinovec et al., 2015) at the MAJOR station.

Particle Number (PN) concentrations were measured with ultrafine Condensation Particle Counters (CPC) on the size range

of 2.5 nm to 3 µm at PEB (CPC 3776, TSI) and from 7 to 2 500 nm at MAJOR (Envi CPC 200, Palas) with a temporal resolution of 1 s. The aerosol number size distribution was measured at both sites (i) in the range 15-660 nm using Scanning Mobility Particle Sizers (SMPS 3936, L-DMA, CPC, TSI) with a scan time of 2 minutes for 105 channels and (ii) in the range 250 nm to 3.2 µm using an optical particulate counter (OPC model 1.109, Grimm Aerosol Technik) with a scan time of 1 minutes for 31 channels. Particle mass concentration ($PM_1$, $PM_{2.5}$, $PM_{10}$) were also estimated by the OPC.

### 2.2.2. Gas phase measurements

Non-Methane Volatile Organic Compounds (NMVOC) were monitored with a proton-transfer-reaction time-of-flight mass spectrometer (PTR-ToF-MS 8000, Ionicon Analytik, Austria) at a 10 s time resolution. Overview of the PTR-ToF-MS operation and data analysis can be found in Marques et al. (2022). The main organic molecules detected during the measurement period are listed in Table S3. Carbon dioxide ($CO_2$), carbon monoxide (CO), methane ($CH_4$) and ammonia ($NH_3$)

were measured at a temporal resolution of 5 s by a cavity ring-down spectrometer (model G2103 for $NH_3$ and model G2401 for $CO_2/CO/CH_4/H_2O$, Picarro) (Martin et al., 2016). Concentrations of regulatory gaseous pollutants were measured at 10 s time resolution with a chemiluminescence analyser (model 200E, Teledyne API) for the combined measurement of nitrogen oxide ($NO_X$, NO and $NO_2$), an absorption spectrometry monitor (model 400E, Teledyne API) for ozone ($O_3$) and a fluorescence analyser (model AF22 Environment SA at PEB station and model 100E Teledyne API at MAJOR station) for sulphur dioxide

($SO_2$).

### 2.2.3. Automatic Identification System DATA

AIS data records from all ships during the measurement period (31 May - 3 July 2021) that were within a 10 km x 10 km area surrounding the port of Marseille were purchased commercially from (MarineTraffic, 2022). The requested AIS database contained 326,590 records, each containing the following information: the vessel's identification number (Maritime Mobile



Service Identity, MMSI), position (latitude and longitude), date and time, status, heading and course angles, speed as well as the last and next ports visited. To improve the quality of AIS data, dataset was i) pre-processed to exclude sailing vessels and pleasure crafts and to remove data redundancy and noise and ii) interpolated at a time step of 90 s to remove trajectory outliers and recover lost AIS data using the PyVT tool developed by Li et al., (2023) and iii) cross-referenced with ship arrivals and departures data supplied by GPMM. Additional vessel parameters such as name, category, year built of vessel and engine and engine power (in kW) were retrieved from the MMSI provided in the database.

## 2.3. Data analysis

Data processing included calibration and validation using internal analyser parameters, intercomparisons, and user interventions, as well as peaks synchronization to compensate for potential variations in analyser response times.

### 2.3.1. Plumes identification and ship assignment

Plume identification was achieved by cross-referencing measurement data with meteorological and AIS data (Ausmeel et al., 2019; Celik et al., 2020; Eger et al., 2023; Krause et al., 2023). The selection steps and criteria used are detailed below.

1. Plume pre-selection using four typical tracers of ship activity ($CO_2$, $NO_X$, BC, PN) and $O_3$.
    i. Calculation of atmospheric background using a lowpass filtered time series in the form of a rolling median with 60 min window size for each of the five selected pollutants. This lowpass describes the variability in the background concentration due to atmospheric physico-chemical processes and regional transport of pollutants but excludes the short-term variation caused by passing ships (Krause et al., 2023).
    ii. Subtraction of the previously calculated background from the raw signal for each of the five selected pollutants.
    iii. Selection of peaks based on concentration variations exceeding 3 times the average of rolling standard deviation ($\sigma$) of the background with 60 min window size. This variation can be negative (i.e. $O_3$ is consumed in the plume) or positive (the other species).
    iv. Plumes for which a peak was identified for at least 3 of the 5 selected pollutants (70% of measured pollutants in the occasional absence of measurements for one or more pollutants) were retained.
2. Only plumes that could be positively attributed to a single ship or a ship category were selected for further analysis. To this end, each plume, ship location and movement data were cross-referenced with wind speed and wind direction according to the methodology outlined in Figure 2. Step i essentially addresses the specific case of low wind speed. For wind speed under 1.5 m.s$^{-1}$, diffusion-induced dispersion may be significant compared to the advection-induced-dispersion (Arya, 1995; Jeong et al., 2013; Rakesh et al., 2019). Under these conditions, it is therefore possible to capture the plume, provided it passes close to the station. Steps ii to v use dispersion cones to narrow the ship search area (steps ii and iv for moving vessels in port and steps iii and v for vessels at berth). The wind direction used for these cones is the average of the wind directions plus or minus 15 degrees over the target period at the measuring station.
3. In a final step, fine tuning of the database was applied, with the following additional criteria:
    i. Plumes that could not be individualised by a return to baseline level were removed.
    ii. Plumes with a duration of less than 1 min were disregarded.



  iii. Plumes with a residence time > 30 min were removed due to attribution uncertainty.

220  iv. Some plumes (12%) from numerous pleasure crafts and passenger shuttles arriving at or leaving the Vieux-Port
    marina (located in Figure S2) were manually recorded in the database under south-westerly wind conditions that
    placed the MAJOR station downwind of these emissions. This was done despite the impossibility of distinguishing
    the plumes individually.

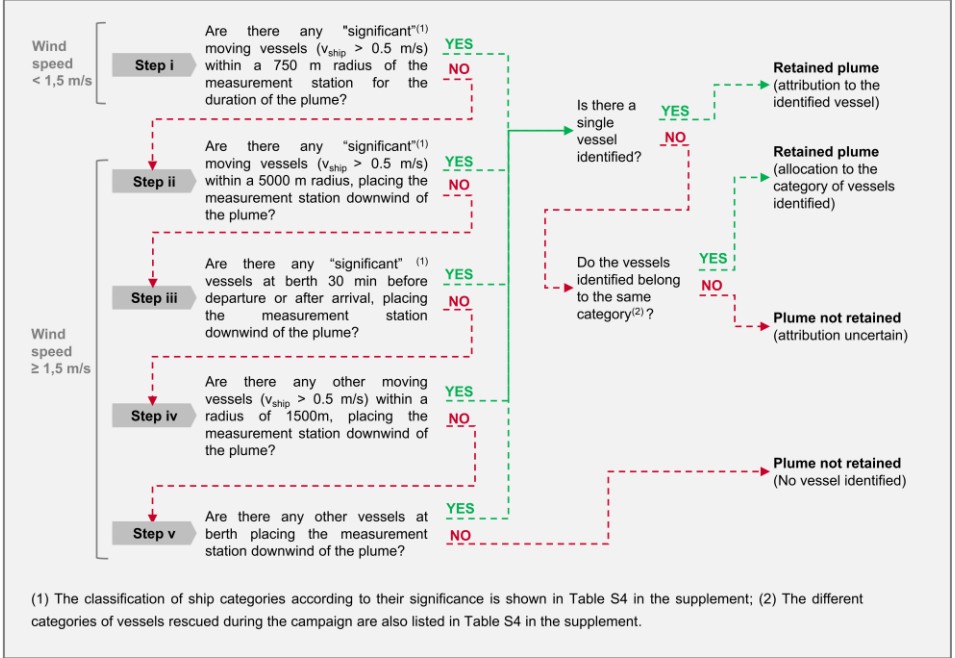

225 **Figure 2. Methodology used for assigning plumes to ships based on AIS and meteorological data.**

### 2.3.2. Quantification of plume characteristics

The most common approach for characterizing the chemical composition of ship plumes is that of emission factors (Ausmeel
et al., 2019; Celik et al., 2020; Pirjola et al., 2014; Van Roy et al., 2022). This approach, described in detail by Celik et al.
(2020), is based on the carbon balance method, which rules out plume dilution and background contribution. For each plume
230 that was successfully assigned to a single ship or a ship category, the emission factor of each pollutant X (EF$_X$), expressed in
grams or particle number per kilogram of fuel used, was derived from Eq. (1) (Diesch et al., 2013).

$$EF_X = \frac{\int_S^E [X]\,(t)\,.\,dt}{\left(\int_S^E [CO_2](t)\,.\,dt\right).\frac{M_C}{M_{CO_2}}} \cdot w_c \qquad (1)$$

where $w_c$ represents the carbon mass fraction in fuel used (fixed at 0.865 kgC.kg$_{fuel}^{-1}$ (Celik et al., 2020; Grigoriadis et al.,
2021b)) ; $M_C/M_{CO_2}$ is the mass fraction of carbon in $CO_2$ ; [X] is the excess (above the atmospheric background) concentration
for pollutant X in µg.m$^{-3}$ for mass concentrations and in $10^{12}$ part.cm$^{-3}$ for number concentration ; [$CO_2$] is the excess $CO_2$
235 concentration in mg.m$^{-3}$ ; and indices S and E denote the start and end of the plume, respectively.





Plume start and end dates (indices S and E) were retrieved from the point of inflection of the concentration to time curve, i.e. when the derivative crosses zero, to avoid any subjectivity. EF were calculated separately for each pollutant measured accounting the response times of each instruments (Ježek et al., 2015). Then, the background to be subtracted from the signal, to determine excess concentrations, was defined by the mean concentrations of two 30-second background intervals before
and after the peak. A toolkit has been developed in the Igor Pro 8 environment (WaveMetrics, USA) to systematically perform these calculations.

Furthermore, particle size distributions in ship emission plumes were computed applying the same method on SMPS measurements. However, the start and end times for defining the background and plume intervals were set identically for all size classes, based on those defined for the CPC analyser. This is because the CPC analyser counts the total number of particles regardless of size and provides a more accurate time resolution (10 seconds). As the plume duration can approach the 2 min
scan time of the SMPS, additional checks were applied. For each plume, the total number of particles measured by the SMPS and the CPC were compared. Only plumes with a Pearson correlation coefficient greater than 0.7 were selected for the analysis.

### 2.3.3. Short-term impact of shipping plumes on ambient air concentration levels

The short-term impact of shipping plumes on ambient air concentrations levels was assessed for each single plume using
Eq. (2).

$$\overline{[X]} = \frac{1}{E - S} \int_{S}^{E} [X](t) \, . \, dt \qquad (2)$$

with ($\overline{[X]}$) the average excess concentrations and $[X](t)$ the excess concentrations over the plume duration period (E-S). Plume start and end dates, as well as the background corrections have been determined using the method described in section 2.3.2. This approach has certain limitations. It excludes a significant number of ship plumes from the analysis, particularly during periods of heavy maritime traffic, which could skew the estimated additional average concentrations. However, this exclusion
ensures that the results are not biased by non-ship sources resembling ships emissions (Eger et al., 2023).

## 3.   Results and discussion

### 3.1.  Campaign overview

### 3.1.1. Meteorological and ship traffic conditions

Plume detection is based on both meteorological and ship traffic conditions. The meteorological conditions observed during
the campaign highlight the complexity of air mass circulation in the study area. According to Figure S1, wind directions vary considerably within the port area, especially during land and sea breezes, which are common at this time of year. Wind direction analysis during the campaign enabled to estimate the probability of the measurement stations being downwind of Marseille



Port's main ship emission areas (located in Figure S2). Combining all areas, this probability was 35% at the PEB station and 20% at the MAJOR station. Detailed probabilities for each zone are detailed in Table S5.

In June 2021, approximately 800 ships arrivals and departures were recorded by the GPMM. That excludes pilot boats which systematically escort vessels from the lane entrance to the mooring berth and back, as well as pleasure crafts and passenger shuttles which mainly access to the marinas of Vieux Port, Estaque, and the Frioul Islands (located in Figure S2) located outside the GPMM sector. Most ships operating in the GPMM were dedicated to passenger transport (40%), including ferries and cruises (35% and 5%, respectively). Cargo ships represent 25% of the activities, while tugs and tanker accounted for 20%

and 10% of port movements, respectively. Other vessels, primarily used for sea rescue, made up the remaining 5%. On average, ship arrivals peaked early in the morning from 04:00 to 06:00 UTC, and departures were most frequent late in the afternoon from 16:00 to 18:00 UTC. This schedule varies slightly according to ship category, as shown in Figure S3.

### 3.1.2. Concentrations and impact of port activities

A total of almost 110 chemical components were measured during the campaign, including 45 metal trace elements (or metals)

and 41 NMVOC. Main statistics of pollutants concentrations measured at both stations PEB and MAJOR are reported in Table S6. Time series of the key substances and the particle size distribution over the whole campaign are given in Figure S4 and Figure S5.

To investigate the impact of port activities on local air quality, concentration levels of pollutants measured simultaneously at the two stations were compared with those from the MRS-LCP station (Table S6), part of the regional air quality network.

This monitoring station, located in the city's center (Figure 1), was chosen because it serves as a reference for urban background pollution (Chazeau et al., 2021). Average concentrations of PN, $PM_{2.5}$, $PM_1$, and NOx at PEB and MAJOR were 1.5 to 2 times higher than those measured at MRS-LCP, indicating a significant influence from port activities. In contrast, average levels of other compounds were similar across all stations. Additionally, analysis of maximum concentrations and the 75th percentile reveals that pollutants such as $SO_2$ and some metals (As, Cd, Co, Fe, Ni, Sb, Se, Sn, V, Zn, and Zr) are 2 to 10

times higher near the port than downtown.

To identify the sources responsible for the high concentrations in the port area, the conditional bivariate probability function (CBPF) (Uria-Tellaetxe and Carslaw, 2014) was computed for measured species. The CPBF is a graphical method in polar coordinates commonly used to highlight wind speeds and directions associated with high concentrations of a pollutant in order to identify emission sources (Adotey et al., 2022; Ryder et al., 2020; Toscano et al., 2022). It estimates the probability that

measured concentrations exceed a predetermined threshold (in this case the 80th percentile) for a given range of wind sector and wind speed. As shown Figure 3, CPBF indicates that the highest concentrations typically occur when the measurement sites are downwind of the mooring berths or the ships' access lanes to the port, except for OA, which also had high concentrations associated with land breeze (to and from the city). These findings are supported by analysis of the daily profiles of concentrations (Figure S6) showing a correlation with ship arrival and departure profiles (Figure S3) depending on the

pollutants.





**Figure 3. Conditional bivariate probability function (CBPF) computed for (a) NO, (b) PN, (c) BC, (d) V, (e) Ni, (f) toluene (NMVOC), (g) SO₂, (h) SO₄²⁻ and (i) OA during the campaign at the PEB and MAJOR stations. The radial axis indicates wind speed in m.s⁻¹, and the colour bar indicates the probability of a species being above the 80th percentile of the compound. Maps taken from © OpenStreetMap contributors 2023 © CARTO. Distributed under the Open Data Commons Open Database License (ODbL) v1.0.**






This correlation is particularly strong for NO, BC, PN, Vanadium (V), and Nickel (Ni). The increased dispersion of concentrations, maxima, and the differences between mean and median concentrations during ship movements underscore the significant impact of their emissions on the concentrations measured in the port. For the other substances such as $SO_2$ and NMVOCs (e.g. toluene) in the gaseous phase, and $SO_4^{2-}$ or OA in the particulate phase, correlations exist but in a lesser extent.

For OA and NMVOCs (e.g. toluene), average concentrations slightly increase during ship movements but rise more notably at night (with greater dispersion and extrema) due to the nighttime land breeze regime bringing urban emissions back to the measurement station. In the case of $SO_2$, average concentrations and peaks also slightly increase during ship movements but enhanced over the morning and longer than for other compounds, suggesting contributions from other sources. This could be due to sea breezes lifting emissions from the large industrial areas located 25 km north-west of Marseille, previously pushed

away to sea by nighttime land breezes (Chazeau et al., 2021). Unlike $SO_2$, $SO_4^{2-}$ does not follow the same daily cycle, showing the concentration increasing in the afternoon, mainly driven by its photochemical formation cycle. It is worth noting that the reduction in sulphur content in fuels in 2020 could consequence lower contributions from ships compared to other sources. Indeed, as shown Figure 4b by the flat pattern of the daily profile observed after the new regulations, $SO_2$ may become a less effective tracer of ship emissions.

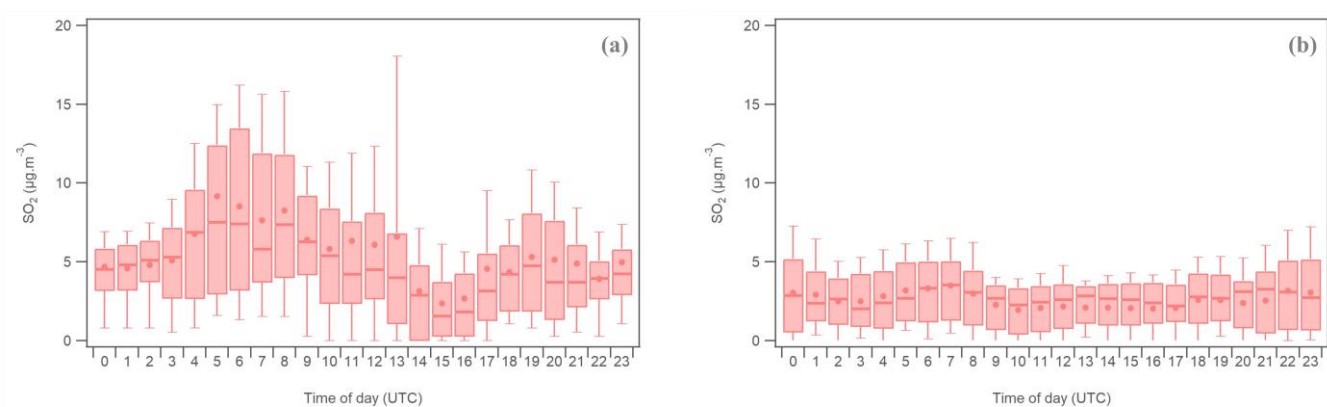


**Figure 4. Daily profiles of $SO_2$ at the PEB station (a) before implementation of the new regulations concerning sulphur content in ship fuels (exploratory campaign to define the location of stations - September 2019) (b) after implementation of these regulations (campaign conducted for this study - June 2021). For each box plot, the coloured box represents the interval between the 25th percentile and the 75th percentile, the vertical lines represent the interval between the 10th percentile and the 90th percentile,**
**whereas the horizontal line and circle are the median and mean, respectively.**

### 3.1.3.Ship plumes

From the measurements taken at the two stations (PEB and MAJOR) and following the procedure described in section 2.3.1, 118 plumes were attributed to ships at berth and 235 plumes to ships "in manoeuvring" (ships at speeds below 2.5 m.s$^{-1}$ (5 knots) or "in navigation" (ships at speeds between 2.5 and 14 m.s$^{-1}$) in or near the port area (less than 750 m from the coast).

The yield of plumes attributed to ships "in manoeuvring" or/and "in navigation" is 29% of the number of passages recorded by the AIS data (see section 3.1.1). This relatively low yield is primarily due to the dependence on wind direction at the time of the ship's passage. The plume can only be captured when the wind direction positions the measurement station downwind





from the ship's emissions. This dependency, combined with the stringent criteria of the plume assignment algorithm, resulted in the exclusion of many plumes, particularly during periods of heavy ship traffic.

The plume sample spans 8 different ship categories, among which are the three main categories operating at GPMM (ro-ro ferries, cargo and cruise ships). However, plume frequency varies by category, with ro-ro ferries comparable to cruise ships, together accounting for 70% of the sample plumes. The remaining categories, in descending order of frequency, include pleasure crafts, cargos, tankers, passenger shuttles, tugs, and rescue vessels. The sample also encompasses the various operational phases observed in a port: at berth (33%), in manoeuvring (16%) and in navigation (51%) with ships generally

maintaining speeds below 5 m.s$^{-1}$ when not docked due to regulatory speed limits in port. This reduced speed generally corresponds to an engine load of less than 25% (Jeong et al., 2023; Knudsen et al., 2022; Lack and Corbett, 2012). However, the breakdown of operational phases varies by ship category. For cruise ships, the "at berth" phase is most common, accounting for 90% of their time. In contrast, the "in navigation" phase predominates for most other categories, at 80%, except for tankers, which show an even distribution between "in navigation" and "in manoeuvring".

Figure S7 shows the example of two successive plumes from different ro-ro ferries arriving at the port. As expected, these emissions mainly consist of a blend of gaseous compounds (such as $CO_2$, $CO$, $NO_X$, $SO_2$) and particulate compounds (BC, $SO_4^{2-}$, OA), accompanied by a decrease in $O_3$ levels. The particle size distribution from these ships shows a bimodal pattern (20 and 80 nm) for the first ship, with sulphur predominantly emitted as $SO_4^{2-}$, and an unimodal pattern (20 nm) for the second ship, where sulphur is mainly as $SO_2$. Regarding metals, the 30-minute time resolution of the analyser prevents distinguishing

emissions between the two ships but reveals the presence of V, Ni, calcium (Ca), and iron (Fe). In terms of NMVOCs, only fragments of unspecified hydrocarbons are detectable.

### 3.2. Emission factors

The methodology described in Section 2.3.2 for calculating EF from the 353 identified ship plumes was applied to all pollutants except trace metal elements. The 30-minute time step necessary to maintain acceptable detection limits is not suitable with the

actual duration of the plumes, which typically range from 2 to 14 minutes. As depicted in Figure S7c, this mismatch results in a single-point spike above background levels for each detected plume, unlike the continuous measurement of other compounds. Furthermore, the relatively low $\Delta CO_2$ values resulting from this time step significantly increase the uncertainties in EF calculations. Sensitivity tests conducted on $NO_X$ and PM for different temporal resolutions—from 10 seconds to 5 minutes— revealed that the median relative deviation from the finest resolution values is less than 10% for resolutions under 2 minutes,

30% for a 2-minute resolution, and greater than 80% for a 5-minute resolution. For temporal resolutions greater than 1 minute, the median deviation increases as plume duration decreases, as shown in Table S7.

In the following subsections, we compare EFs obtained in this study with both those reported in the literature (as summarized in Table S8, which includes data from over 30 studies using various experimental methods) and those reported in the regional air quality monitoring network's emissions inventory used to model the atmospheric dispersion of ship emissions.

Additionally, we investigate how ship-related characteristics— ship category, engine power, engine age, operating phase, ship



speed and plume age—affect gaseous and particulate emissions, as well as particle size distribution, during port operations in Marseille.

Due to the non-normal distribution of emission factors, EF values are consistently reported as medians with interquartile ranges [25th–75th percentiles]. For the same reason, statistical tests for significance in group comparisons are conducted using the Kruskal-Wallis test, followed by post-hoc Dunn tests with Bonferroni correction when the groups do not share the same central tendency (Borge et al., 2022; Marmett et al., 2023).

To improve the readability of the results and graphs, specific ship characteristics have been grouped. For operational phases, "manoeuvring" and "navigation" have been combined into a single group labelled "manoeuvring/navigation" due to the similar emission factors observed for most pollutants in these phases. Regarding vessel categories, tankers, passenger shuttles, pilot boats, tugs, and rescue vessels have been grouped under "other categories" because of the limited variety and/or small number of plumes identified for these vessels. Pleasure crafts were also included in this group to ensure consistency in the characterization of gaseous and particulate phases. However, since pleasure crafts were only identified at the MAJOR station, no chemical characterization of the particulate phase could have been conducted for this category. Statistically significant differences within these groups are discussed in the main text.

In the remainder of the article, each box plot is presented with a coloured box representing the range between the 25th and 75th percentiles, while the vertical lines denote the interval between the 10th and 90th percentiles. The horizontal line and circle indicate the median and mean, respectively, and the grey dots represent the extremes. Additionally, a table associated with each box plot provides the Number of Studied Plumes (NSP), the Number of Quantified Plumes (NQP), the Total Duration of Quantified Plumes in hours (TDQP), and the Number of Different Vessels in the Quantified Plumes (NDVQP).

**3.2.1.Gaseous phase**

Figure 5 shows the distribution of EFs across all gaseous compounds (see Table S10 for detailed statistics). NMVOCs and NH$_3$ are excluded from these plots because their median and percentile values are below the detection limits, except for aromatic C8 compounds and toluene. These exceptions are discussed in more detail below. The large variability of EFs observed is consistent with findings in the literature (Table S8) and is further accentuated when EFs are compared without considering the specific characteristics of the ships or the fuels used.





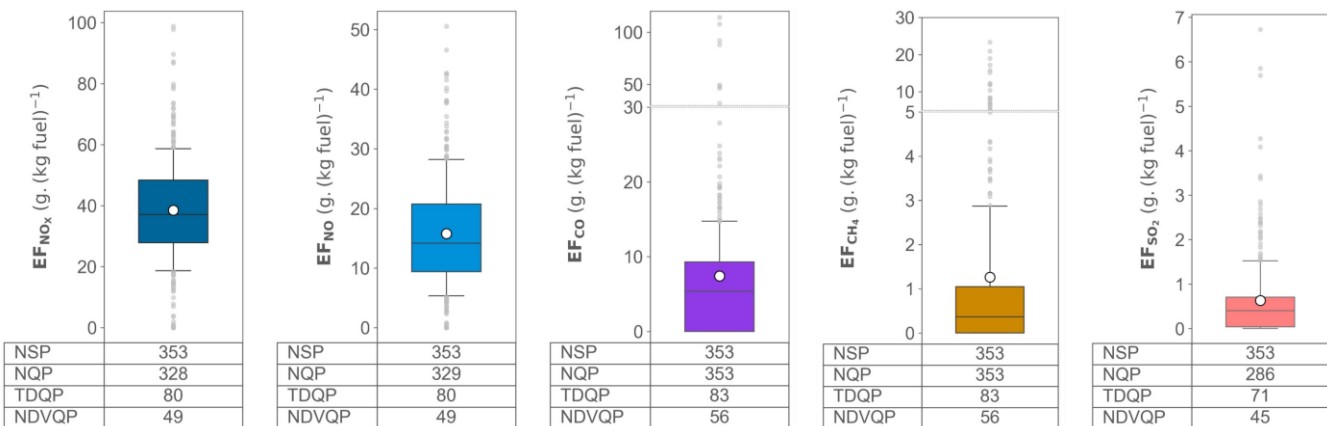

**Figure 5. Distribution of EFs for gaseous compounds across all identified plumes.**

**Nitrogen oxides (NO$_X$)**

For NO$_X$, the median EF of the plumes studied (37 g.kg$_{fuel}$$^{-1}$ [28-48]) is comparable to the low range of values reported in the literature for ships using fuel oil (average of $57 \pm 26$ g.kg$_{fuel}$$^{-1}$) and 2 times lower than the value taken into account in regional emissions (80 g.kg$_{fuel}$$^{-1}$) inventory, which lies in the high range of values in the literature. Only 2% of the 353 EFs determined from ship plumes exceed this value.

The median EF$_{NO}$ (14 g.kg$_{fuel}$$^{-1}$ [9-16]) is significantly lower than the values reported in the literature (average of

$70 \pm 34$ g.kg$_{fuel}$$^{-1}$) as most of these values stem from measurements taken directly from the exhaust of ships' engines, where NO$_X$ emissions are more than 90% NO (Zhao et al., 2020). This balance is quickly altered in the atmosphere, where NO oxidises rapidly to NO$_2$. In this case, the oxidation reaction is driven by ozone (O$_3$) because the initial NO concentrations in the exhaust gases are much higher than the ambient O$_3$ concentrations, which leads to a local reduction in O$_3$ concentrations and a decrease in the NO/NO$_X$ ratio (see Figure S7a). For this reason, most studies explicitly report only the EF$_{NO}$.

Nevertheless, field campaign studies that take this oxidation into account report EF$_{NO}$ of the same order of magnitude as those determined in this study ($7 \pm 1$ g.kg$_{fuel}$$^{-1}$ (Celik et al., 2020) and $16 \pm 11$ g.kg$_{fuel}$$^{-1}$ (Diesch et al., 2013)).

Our results showed that the main influence factor is the plume age (or residence time) and in a lesser extend the ship category. Beside these factors, other studies also pointed out operational phase, ship speed/engine load, as well as engine (Celik et al., 2020; Grigoriadis et al., 2021a; Huang et al., 2018; Peng et al., 2020; Sugrue et al., 2022).

As shown in Figure 6, as plume age increases, both EF$_{NOx}$ and NO/NO$_X$ ratio decrease during daytime, while remaining stable during nighttime. During daytime, the NO/NO$_X$, decreases rapidly from 0.9 at emission (Zhao et al., 2020) to 0.4 for the youngest plumes and 0.3 for plumes older than 15 minutes. This latter value is close to the one corresponding to photochemical equilibrium (0.2) suggested by Celik et al (2020) for plumes older than 30 minutes. In addition, the diurnal two-fold decrease in EF$_{NOx}$ between the shortest and longest plume age suggests the existence of NO$_X$ sinks involving photochemical reactions

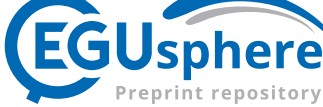

with radicals such as OH, HO$_2$ and RO$_2$ or with NMVOCs (Celik et al., 2020). These reactions could lead to the formation of

nitric acid and, through heterogeneous reactions, to the production of aerosols containing nitrates or organo-nitrates.

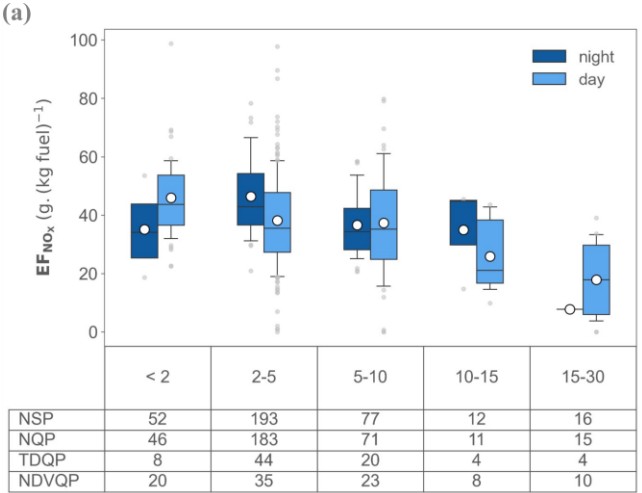
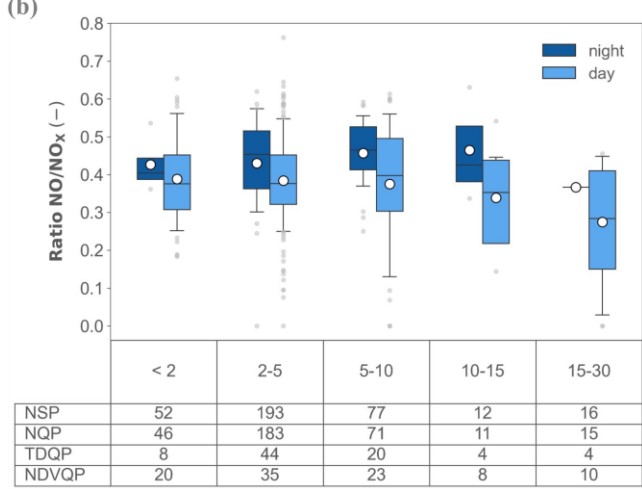

**Figure 6. Distribution of (a) EF$_{NOx}$ and (b) NO/NO$_X$ ratios as a function of plume age expressed in minutes and time of day (daytime and nighttime).**

In addition, as shown in Figure 7, the categories grouped under "other categories" exhibit lower EF$_{NOx}$ compared to cruise

ships, cargos and ferries. Specifically, the two categories within this group, "pleasure craft" and "passenger shuttles," have the

lowest emissions: EF$_{NOx}$ are 1.3 times lower than those of all the other categories combined (31 g.kg$_{fuel}^{-1}$ [23-35] versus

40 g.kg$_{fuel}^{-1}$ [32-51]). This difference is likely due to the low engine power of these ships. (Sinha et al., 2003) or the type of

fuel used (petrol can be used for pleasure craft). This figure also suggests that the operational phases do not significantly affect

EFs and confirms that EF$_{NOx}$ for vessels "at berth" or operating with an engine load below 30% are of a similar magnitude

(Grigoriadis et al., 2021a). It should be noted that for ferries "at berth" , the observed EFs are lower due to the limited number

of plumes identified, coupled with a higher proportion of plumes older than 10 minutes detected during the day.

Finally, analysis of engine age retrieved from the AIS ship tracking data, indicate that the EF$_{NO_X}$ are not affected by the Tier

regulations on NO$_X$ production imposed by the MARPOL convention (Tier 0 engines built before 2000, Tier I before 2011,

Tier II before 2016 and Tier III after 2016). This result suggests, as previously pointed out by Knudsen et al (2022) and Sugrue

et al (2022), that these regulations have little influence on NO$_X$ emissions, particularly in the case of low engine loads and for

Tier 0 and Tier 1 ship categories (the most represented categories (90% of the plumes studied for which the engine age was

known)).





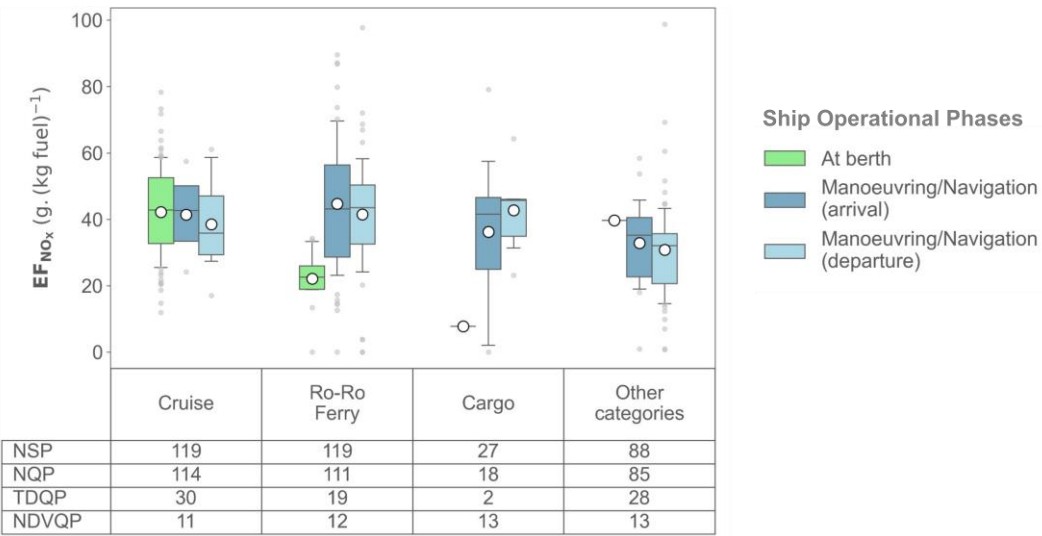

**Figure 7. Distribution of NO$_X$ emission factor as a function of ship category and operational phase.**

**Sulphur dioxide (SO$_2$)**

The median SO$_2$ emission factor of the plumes studied (0.4 g.kg$_{fuel}^{-1}$ [<0.1-0.7]) is comparable to the average value reported in the literature for ships using fuel oil with sulphur contents < 0.1% (1.1 ± 1.0 g.kg$_{fuel}^{-1}$). It is nevertheless more than 5 times

lower than the value provided by the regional emissions inventory (2.0 g.kg$_{fuel}^{-1}$) for the same fuel sulphur content category. Analysis of the parameters likely to influence SO$_2$ EFs reveals that its emissions depend mainly on ship category and operational phase. As shown in Figure 8, ships equipped with engines of more than 10,000 kW (cruisers, ferries and cargo ships) emit more SO$_2$ than ships with engines of less than 4,000 kW (other categories). In addition, ships "at berth" using their auxiliary engines generate less SO$_2$ than ships in "manoeuvring/navigation" using mainly their main engine. Regarding the

"manoeuvring/navigation" phases, a notable distinction in SO$_2$ emissions is observed between arrivals and departures, with higher emissions during arrivals. This distinction is more pronounced when the number of plumes identified in these phases is high, as is the case for ferries. For cruises, cargo ships and tankers (included in "other categories"), the same trend is observed, although the number of plumes on arrival and departure is lower. The difference in SO$_2$ emissions between arrivals and departures could reflect the transitional period when the ship makes the required fuel change in the Marseilles harbour

(switching from fuels with a sulphur content of 0.5% to 0.1% in response to regulations) and/or be linked to the use of open-loop scrubbers, which are required to be shut down in the Marseilles harbour. When this type of scrubber is shut down, a temporary increase in SO$_2$ emissions can be observed (Teinilä et al., 2018).





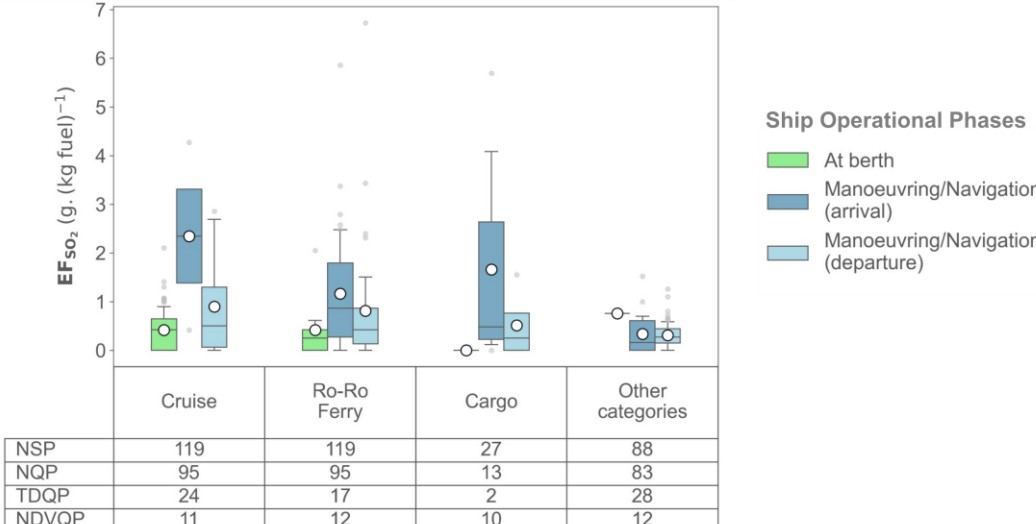

**Figure 8. Distribution of SO₂ emission factor as a function of ship category and operational phase.**

In conclusion, as highlighted by numerous studies (Grigoriadis et al., 2021a; Huang et al., 2018; Zhao et al., 2020) $SO_2$ emissions depend mainly on the sulphur content of the fuel used. During the campaign, all the plumes from ships "at berth" had sulphur contents < 0.1%. For ships in "manoeuvring/navigation", sulphur levels were systematically < 0.5% and only 10% of the plumes measured had sulphur levels above 0.1%, mainly including incoming ships.

**Carbon monoxide (CO)**

The median CO emission factors ($EF_{CO}$) for the plumes studied (5.4 g.kg$_{fuel}^{-1}$ [<1.3-9.3]) is comparable to the values reported in the literature for ships using fuel with a sulphur content of <0.1% (average of 5.7 ± 8.6 g.kg$_{fuel}^{-1}$) as well as to the value used in regional emissions inventory (7.5 g.kg$_{fuel}^{-1}$) for this type of fuel.

Analysis of the parameters likely to influence $EF_{CO}$ reveals that CO emissions depend mainly on the operational phase (Figure
9). Ships "at berth" emit less CO than ships in "manoeuvring/navigation". As fuel type has little or no influence on CO emissions (Petzold et al., 2011) the variation of $EF_{CO}$ within the operational phases is attributable to engine load. An increase in engine load leads to a rise in combustion temperature, making it more efficient and thus reducing CO (Agrawal et al., 2010; Zetterdahl et al., 2016). Ships "at berth", which mainly use their auxiliary engines operating at a stable and optimal engine load, therefore emit less CO than ships in "manoeuvring/navigation" using their main engine at lower and less stable loads.

The effect of combustion temperature is also reflected within "manoeuvring/navigation" phase, with higher emissions observed on departure than on arrival, probably due to the "cold" start of the main engines and the resulting incomplete combustion conditions. In addition, plumes with particularly high emission factors (30-100 g.kg$_{fuel}^{-1}$) has been observed likely corresponding to changes in engine speed during acceleration or deceleration phases (Bai et al., 2020; Huang et al., 2018; Jiang et al., 2021) as these plumes are systematically captured at the port exit.





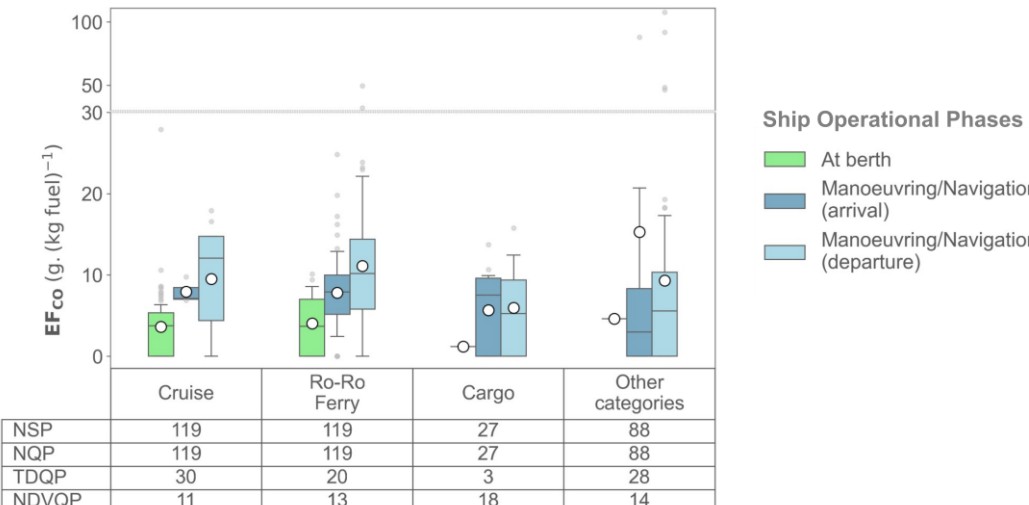

**Figure 9. Distribution of CO emission factor as a function of ship category and operational phase.**

**Methane (CH₄)**

The methane median EFs of the plumes studied (0.4 $g.kg_{fuel}^{-1}$) is comparable to the EFs reported in the literature for ships using fuel oil (average 0.2 ± 0.4 $g.kg_{fuel}^{-1}$) and align with the value used in regional emissions inventories (0.3 $g.kg_{fuel}^{-1}$). However, the mean EFs (1.3 ± 0.3 $g.kg_{fuel}^{-1}$) is higher because 10% of the plumes exhibit EFs>1.0 $g.kg_{fuel}^{-1}$, with some reaching up to 23 $g.kg_{fuel}^{-1}$. Analysis of the various ship parameters did not reveal any significant driver for cluster with EFs>1.0 $g.kg_{fuel}^{-1}$. Furthermore, the distribution analysis of $NO_X$ and $SO_2$ EFs does not indicate the use of methane-based fuels, such as Liquefied Natural Gas (LNG) or Gas to Liquids (GTL), which are typically associated with higher $CH_4$ EFs but lower $NO_X$ and $SO_2$ EFs. For further details on the other hypotheses considered, based on our knowledge of the study area and which could potentially be combined to explain the higher observed EFs, please refer to Table S12.

**Non-methane volatile organic compounds (NMVOCs)**

For the majority of NMVOCs, the mean, median and percentile EFs observed with the plumes are below the detection limits (DL), which vary between 5 and 200 $mg.kg_{fuel}^{-1}$ depending on the compounds (Table S10). These results are consistent with EFs reported in the literature, where values derived exclusively from direct emission measurements, are generally below 30 $mg.kg_{fuel}^{-1}$ (Agrawal et al., 2008a, 2010; Huang et al., 2018; Timonen et al., 2022).

However, for certain compounds such as C8 aromatics and toluene, the 90th percentiles exceed the detection limits, suggesting that these compounds may occasionally be emitted in greater quantities. Nevertheless, due to the detection limits and measurement uncertainties for these compounds, it is challenging to reliably determine the parameters associated with higher emissions. It appears nevertheless, that among the parameters examined, the operational phase—and consequently the type of



engines and fuel used—exerts the most significant influence. Indeed, a detailed examination of EFs by operational phase, as shown in Table S11, indicates that:

- for toluene, EFs from ships in "manoeuvring" phase are higher than in other operating phases. Huang et al (2018) and Timonen et al. (2022) have also highlighted this when studying the emissions of ships (Cargo and Ro-ro Ferry) during different operating phases. The study by Huang et al (2018) further showed that, in the "manoeuvring" phase, the quantity of toluene emitted was four times higher when low-sulphur fuel (0.4%) was used compared to when higher-sulphur fuels (1.1%) were used.
- C8 aromatics are emitted in greater quantities during "at berth" and "manoeuvring" phases, a trend also noted by Huang et al (2018) and Timonen et al (2022).

**Ammonia (NH$_3$)**

The ammonia emission factors of the plumes analysed are systematically below the DL of 0.1 g.kg$_{fuel}^{-1}$ except for two plumes with EFs close to this threshold, with values of 0.12 and 0.15 g.kg$_{fuel}^{-1}$. These values are in line with the literature, which reports an average of 0.07 ± 0.14 g.kg$_{fuel}^{-1}$ (across all fuel types).

Figure 10 provides an overview of the median global emission profile of gaseous phases and illustrates its variability according to the operational phases of ships, which most commonly account for the variations in emission factors (EFs) of different compounds. Ship gaseous emissions are primarily composed of NO$_X$ (86%) and CO (12%), while SO$_2$ and CH$_4$ represent each about 1%. Other compounds, such as NMVOCs, constitute less than 0.1% of the gaseous phase but can account for up to 10% under certain operational conditions only identified during ships were "at berth" or "in manoeuvring", which may significantly impact the formation of secondary pollutants.

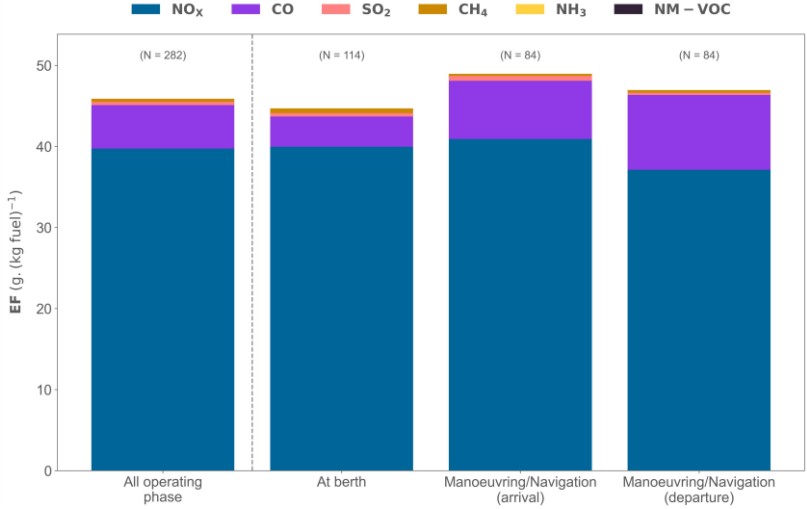

**Figure 10. Median gaseous composition of ship emissions as a function of operational phase. Number of plumes considered (N) is shown for each phase.**



### 3.2.2. Particulate mass vs particle number

$EF_{PN}$ obtained from particle counter measurements (CPC) and $EF_{PM1}$ obtained from particle size measurements (SMPS) are illustrated using box diagram in Figure 11 (for detailed statistical analysis, refer to Table S10).

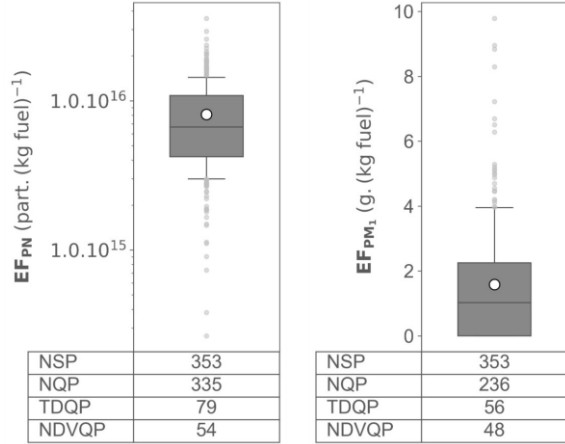

**Figure 11. Distribution of EFs for particulate matter by number (PN) and by mass (PM$_1$) across all identified plumes.**

The median $EF_{PN}$ (6.7 $10^{15}$.part.kg$_{fuel}^{-1}$ [4.2-10.8]) is comparable to the values reported in the literature for ships using fuel oil with sulphur contents <0.1% (mean of 8.1 ± 14.1 $10^{15}$.part.kg$_{fuel}^{-1}$).

Regarding the overall submicronic particle mass (PM$_1$) emission factors, the use of the SMPS analyser was preferred over the OPC for the calculation. Inter-comparisons showed that the OPC could underestimate PM$_1$ concentrations by up to a factor of
3, particularly when the measurement sites were downwind of ship plumes, due to its inability to measure particles smaller than 250 nm in diameter. The median of the plume $EF_{PM1}$ thus obtained (1.0 g.kg$_{fuel}^{-1}$ [<0.4-2.25]) is comparable to the high range of values reported in the literature for ships using fuel oil with sulphur contents < 0.1% (average of 0.6 ± 0.2 g/kg$_{fuel}^{-1}$). Considering the PM$_1$/PM$_{2.5}$ ratio (average 0.8), derived from the OPC measurements, the estimated $EF_{PM1}$ are also in line with $EF_{PM2.5}$ (1.4 g.kg$_{fuel}^{-1}$) considered in regional emissions inventories.

Analysis of the parameters likely to influence $EF_{PN}$ and $EF_{PM1}$ reveals a statistically significant dependence of emissions on the operational phase. As shown in Figure 12, ships "at berth" generate more PN and less PM$_1$ than when they are in "manoeuvring/navigation" (9.9 $10^{15}$.part.kg$_{fuel}^{-1}$ [7.9-12.1] versus 5.2 $10^{15}$.part/kg$_{fuel}$ [3.4-8.1] for $EF_{PN}$ and 0.6 g.kg$_{fuel}^{-1}$ [<0.4-0.9] versus 1.7 g.kg$_{fuel}^{-1}$ [0.8-3.3] for $EF_{PM1}$). It should be noted that no systematic bias related to the age of the plumes has been observed. The distribution of plumes in the different plume age classes is similar for ships "at berth" and those in
"manoeuvring/navigation".



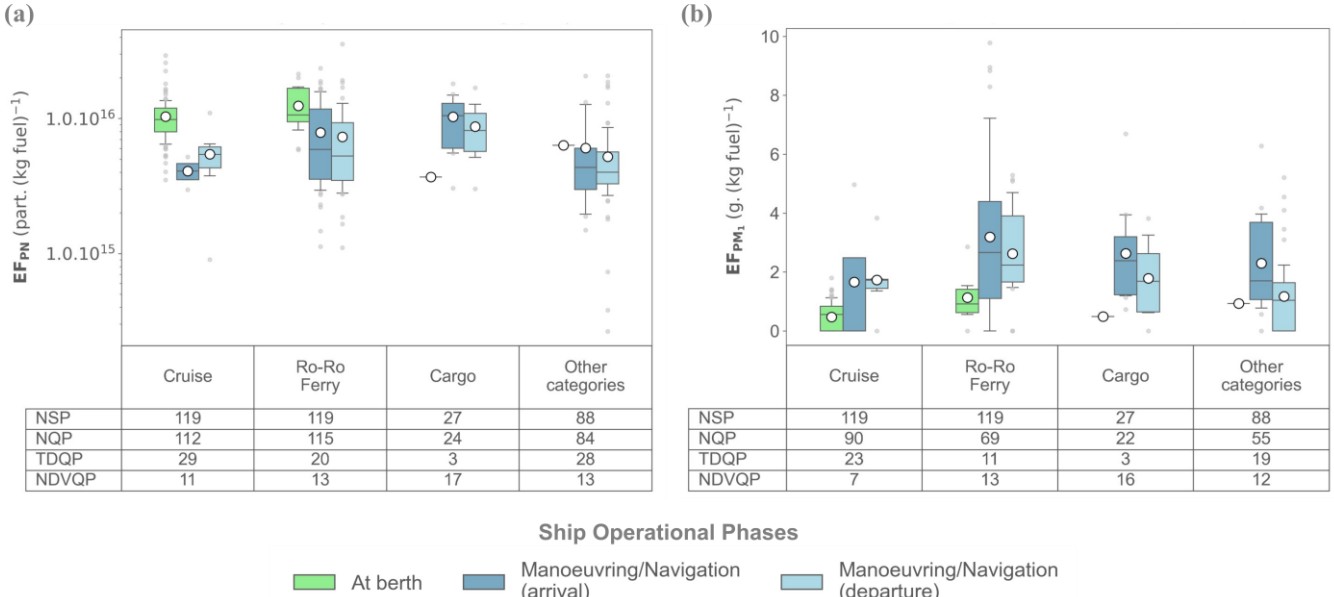

**Figure 12. Distribution of EFs (a) for PN and (b) for PM$_1$ as a function of ship category and operational phase.**

These variations can be explained mainly by the use "at berth" of auxiliary engines operating at a stable and optimum engine speed with low sulphur distillate fuel (< 0.1%), whereas ships in "manoeuvring/navigation" within the port area use their main engine at a lower (< 25%) and less stable engine load. These observations indicate that PN and PM$_1$ emissions are influenced by engine load (Anderson et al., 2015a; Grigoriadis et al., 2021a; Zetterdahl et al., 2016) and engine speed (Diesch et al., 2013). The contrasted evolution of the mass and number of particles observed in Figure 12 between ships "at berth" and "manoeuvring/navigating" has also been highlighted by Anderson et al (2015b) and Chu-Van et al (2018). This unusual development is attributable to the particle size distribution (see section 3.2.4) and not to particle formation. When ships are docked, particles are mainly smaller than 50 nm in diameter, whereas when "manoeuvring/navigation" a mode around 100 nm appears and may even become predominant, thus contributing more to the total mass of PM$_1$ than finer particles.

In addition, as with SO$_2$, a difference was observed between emissions of PN and PM$_1$ during arrivals and departures, with arrivals showing higher emissions (Figure 12). However, for cruise ships, this trend does not hold, but the small number of plumes identified for this category during these operational phases makes this result uncertain. In the analysis of the evolution of the SO$_2$ two hypotheses were considered to explain the differences between arrivals and departures: fuel transition and the shutdown of open-loop scrubbers. The fact that this distinction between arrivals and departures is equally marked for both the number and mass of particles suggests that fuel switching is the most likely hypothesis. Indeed, emissions of PN and PM$_1$ decrease with the sulphur content of the fuel (Celik et al., 2020; Diesch et al., 2013; Grigoriadis et al., 2021a) but also as fuel quality improves (from residual fuel oil to distillate fuel oil) (Gysel et al., 2017; McCaffery et al., 2021).

Finally, Figure 12 also indicates that "manoeuvring/navigation", ferries and cargo ships equipped with engines of more than 10,000 kWh would emit more particles in mass and number than other ships with engines of less than 4,000 kWh. For cruise





ships, which are equipped with engines similar to those of ferries and cargo ships, it is challenging to draw definitive conclusions due to the limited number of plumes identified during these operational phases, especially since the emission factors for cruise ships 'at berth' are comparable to those of ferries."


The age of the plumes also seems to affect the number and/or mass of particles, as shown in Figure 13. Regarding the number of particles (PN), the youngest plumes (less than 5 minutes old for ships "at berth" and less than 2 minutes old for ships "in manoeuvring/navigation") have higher emission factors (a factor of 1.5 and 2 respectively) compared with older plumes. This observation suggests particle accumulation and/or coagulation processes, which reduce the number of particles but increase

their average size (Celik et al., 2020; Lack et al., 2009). For particulate mass (PM$_1$), an upward trend was observed only for ships "at berth". However, the results of the Kruskal-Wallis statistical tests indicate that all the groups show a similar central tendency, signifying a stability of the aerosol mass for the age range of the plumes studied in this study (< 30 min). Thus, the increase in total aerosol mass due to photochemical ageing, observed by several authors using reactors simulating atmospheric oxidation over periods of 2 to 6 days (Lanzafame et al., 2022; Timonen et al., 2022) was not noticeable for plumes less than

30 minutes old. However, the wide variability of PM$_1$ emission factors in the different age classes precludes any definitive conclusion.

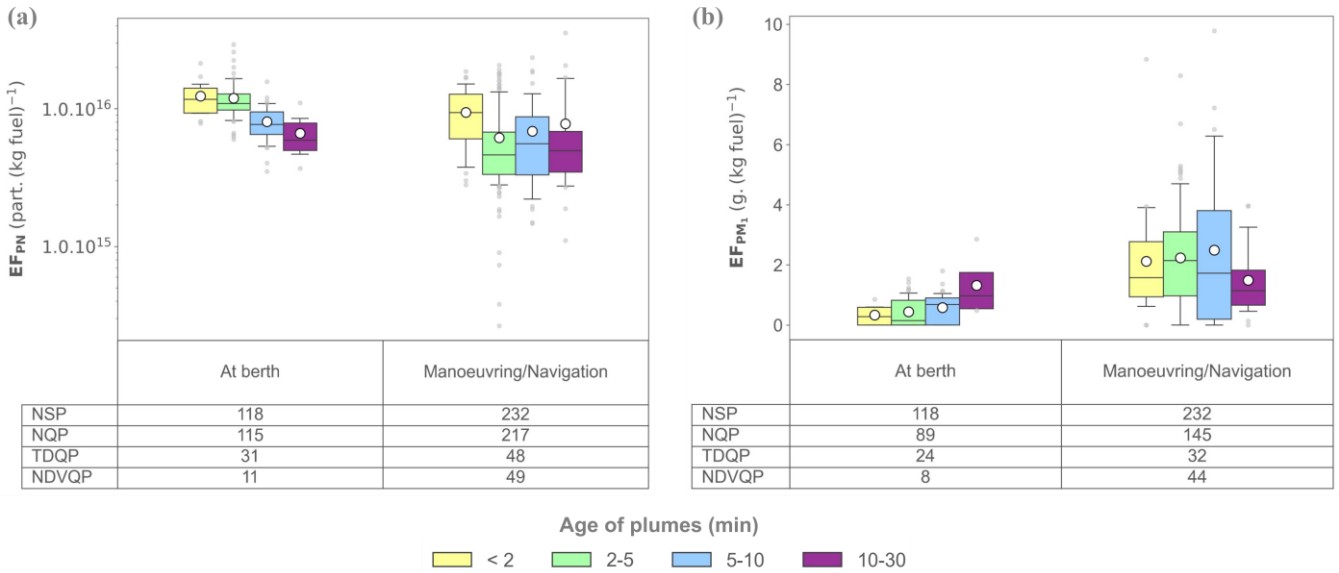

Figure 13. Distribution of EFs (a) for PN and (b) for PM$_1$ as a function of operational phase and plume age.






### 3.2.3. PM$_1$ chemical composition

Figure 14 shows the box plot analysis of the PM$_1$ components EF (for detailed statistical analyses, refer to Table S10). Cl$^-$ is not included in analysis, as its median and percentiles are below the DL.

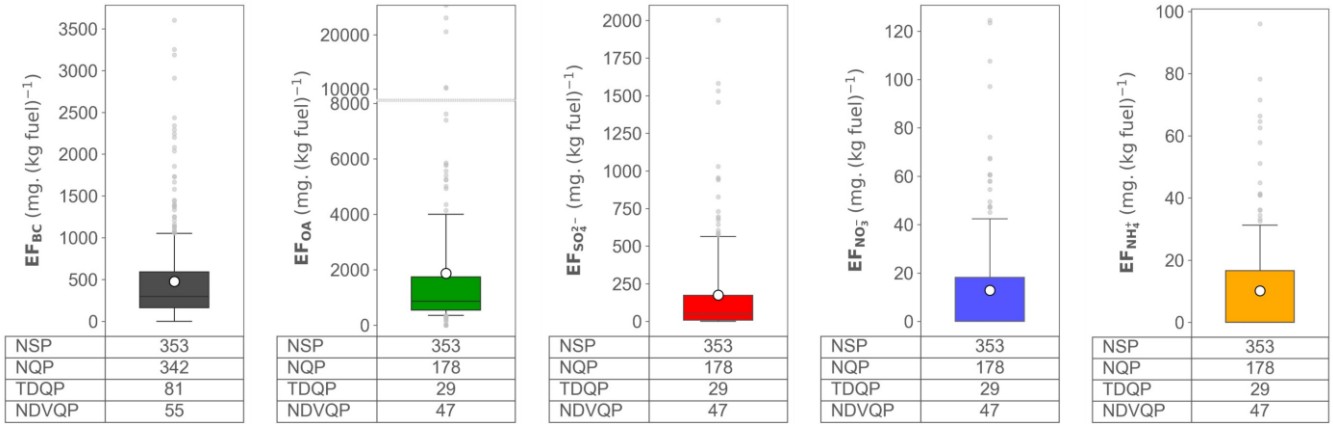

Note: The number of plumes for which compounds were quantified using HR-ToF-AMS analysers (OA, SO$_4^{2-}$, NO$_3^-$, NH$_4^+$) is nearly half that BC due to the exclusive deployment of these analysers at the PEB station.

**Figure 14. Distribution of EFs for PM$_1$ components across all identified plumes.**

The emission factors values determined in this study, are generally comparable to those documented in the literature: for BC, OA and SO$_4^{2-}$ the median EFs are 298 mg.kg$_{fuel}^{-1}$ [163-592], 863 mg.kg$_{fuel}^{-1}$ [543-1,742] and 50 mg.kg$_{fuel}^{-1}$ [<28-174] respectively). These values are comparable to the averages reported in the literature for ships using fuel oil with sulphur contents <0.1% (respectively 238 ± 305 mg.kg$_{fuel}^{-1}$, 624 ± 335 mg.kg$_{fuel}^{-1}$ and 120 ± 50 mg.kg$_{fuel}^{-1}$, Table S8). For NO$_3^-$ and

NH$_4^+$, the EFs median are below the DLs (5.4 and 5.0 mg.kg$_{fuel}^{-1}$ respectively) but remain in agreement with the values reported in the literature for ships using fuel oil with sulphur contents < 0.1% (averages of 3 ± 6 mg.kg$_{fuel}^{-1}$ and 2 ± 3 mg.kg$_{fuel}^{-1}$ respectively, Table S8).

### Black Carbon (BC)

Analysis of the parameters likely to influence BC emissions reveals that they depend mainly on the operational phase. Ships "in manoeuvring/navigation" generate more BC (479 mg.kg$_{fuel}^{-1}$ [261-801]) than ships "at berth" (165 mg.kg$_{fuel}^{-1}$ [105-247]) (Figure 15-a). As for many compounds, this is linked to the low engine load and its reduced stability during navigation and manoeuvres within the port (Sugrue et al., 2022; Zhao et al., 2020). As for SO$_2$, for these operating phases, a distinction is also observed between arrivals and departures, probably due to the change of fuel required when entering the port, switching to a

more refined fuel in response to regulations (Grigoriadis et al., 2021a; Huang et al., 2018; McCaffery et al., 2021). Finally, for this compound, the analysis of EFs as a function of engine age demonstrated the influence of the Tier regulations imposed by the MARPOL Convention on BC production (Tier 0 engines built before 2000, Tier I before 2011, Tier II before 2016 and



Tier III after 2016). A statistically significant decrease was observed (Figure S8) between Tier 0 and Tier I class vessels (the categories most represented (90% of the plumes studied for which the engine age was known)). The same trend holds for the

Tier II and Tier III classes, even if it is not statistically relevant due to the small number of plumes for these categories. This result corroborates those of Sugrue et al (2022) and suggests that these regulations influence BC emissions, even for low engine loads. It is important to note that the distribution of the 4 Tier classes is similar whatever the operational phases and the age of the plumes, thus eliminating any systematic bias that might alter the previous interpretations.

The statistical tests carried out on the other parameters (category of vessel (Figure 15-a) and plume age (Figure 16-a)) indicate

that all groups show a similar central tendency.

Numerous studies have been carried out on the relationship between quantities of BC and CO emitted from various combustion processes. In most cases high correlations were observed (Guo et al., 2017; Taketani et al., 2022; Zhou et al., 2009) as both components arise from incomplete combustion of carbon-based fuels, and the slope of the linear regression ($\Delta BC/\Delta CO$) was often used to identify sources such as petrol/diesel vehicles or biomass combustion (Guo et al., 2017). The correlation between

the $EF_{BC}$ and $EF_{CO}$ of the plumes identified in the present study is negligible ($R^2 < 0.1$) indicating that the emissions of these compounds evolve in an independant way probably due to the use of different fuels. The study by Zhao et al (2020) showed a very strong correlation for a cargo ship and an HFO fuel ($R^2 =0.91$) obtained at different engine loads, but the analysis of the correlation for this same ship with another type of fuel (MDO) shows a less good correlation ($R^2 =0.4$). The correlation becomes negligible ($R^2 < 0.1$) when all the various ships with different fuels (set of EFs compiled in Table S8) are considered.


**Organic Aerosol (OA) and Sulphates ($SO_4^{2-}$)**

OA and $SO_4^{2-}$ emissions also depend mainly on the operational phase. Ships "in manoeuvring/navigation" generate more OA and $SO_4^{2-}$ than ships "at berth" (1,603 mg.kg$_{fuel}$$^{-1}$ [1,095-3,382] versus 611 mg.kg$_{fuel}$$^{-1}$ [470-800] for OA (Figure 15-b) and 171 mg.kg$_{fuel}$$^{-1}$ [55-466] versus <28 mg.kg$_{fuel}$$^{-1}$ [<28-50] for $SO_4^{2-}$ (Figure 15-c)). These variations can be explained mainly by

the use "at berth" of auxiliary engines operating at a stable and optimum engine speed with a distilled fuel with a low sulphur content (< 0.1%), whereas ships "in manoeuvring/navigation" within the port area use their main engine at a lower engine load (< 25%) and less stable with fuels potentially containing a little more sulphur.

For OA, as for BC, a difference was observed between EF on arrival and departure of ships, with arrivals showing emissions 1.3 times higher (Figure 15-b). This distinction probably results from the change in fuel required on entering the port, with a

switch to a more refined fuel (Gysel et al., 2017; McCaffery et al., 2021). However, this phenomenon is not observed for $SO_4^{2-}$ (Figure 15-c). This could be related to Gysel et al (2017) the reduction in $SO_4^{2-}$ emissions due to a reduction in the sulphur content of fuels is negligible, particularly for fuels with an already low sulphur content (Gysel et al (2017).

In addition, when they are "in manoeuvring/navigation", ferries have $EF_{OA}$ twice higher than the other categories (664 mg.kg$_{fuel}$$^{-1}$ [425-1,059] compared with 321 mg.kg$_{fuel}$$^{-1}$ [200-555]) (Figure 15-b) and $EF_{SO4}$ four times higher than the other

categories (358 mg.kg$_{fuel}$$^{-1}$ [195-582] compared to 86 mg.kg$_{fuel}$$^{-1}$ [<28-174]) (Figure 15-c)).



As this observation is not the same for vessels with equivalent engine power, the hypothesis considered is that this vessel category would use a different fuel and/or after-treatment devices than the other categories during navigation and/or manoeuvres. The validity of this hypothesis is strengthened by the fact that some ferries in the port of Marseille are equipped with scrubbers, a notable feature for this category of vessel.

The statistical tests carried out on the age of the OA and sulphate plumes (Figure 16-b,c) indicate that all the groups show a similar central trend.

**Nitrates (NO$_3^-$)**

The emission factors for NO$_3^-$ are often below the DL (more than 60% of measurements), and this remains true even at night.

As a result, statistical analyses of the influence of the various parameters indicate a similar central tendency for all groups, regardless of the parameter analysed. The low nitrate EFs and the absence of any significant variation in NO$_3^-$ as a function of the age of the plume (Figure 16d) do not support the hypothesis put previously forward about NO$_X$ sinks due to photochemical reactions leading to the production of nitrate aerosols. Moreover, according to Celik et al (2020) the high ambient temperatures observed during the measurement campaign limit the presence of this species in the particulate phase.


**Ammonium (NH$_4^+$)**

The emission factors for NH$_4^+$ are often below the DL (more than 60% of measurements). The analysis of ratio between NH$_4^+$$_{mesured}$ and NH$_4^+$$_{predicted}$ (Figure S9)—indicating particle acidity (Zhang et al., 2007)—for ship plume and background conditions suggests that while background particles are fully neutralized, those from ship are not or only partially. The

neutralisation level depends on sulphate emission and consequently on operating phase. When sulphate emissions are high (EF>250 mg.kg$_{fuel}^{-1}$) the near zero slope , similar to that observed by Fossum et al. (2024) suggests that the sulphate measured is mainly in the form of sulphuric acid. For sulphate emission under this threshold, partial neutralisation occurs. In these cases, ammonia concentration levels (3 ppb [2.3-3.7] (Table S6) from city road traffic and agricultural activities are insufficient to neutralise the sulphate emitted by ship or plumes are too young  to reach equilibrium






**Figure 15. Distribution of EFs as a function of ship category and operational phase for (a) BC, (b) OA, (c) sulphate, (d) nitrate and (e) ammonium.**







**Figure 16. Distribution of emission factors (EF) as a function of operational phase and plume age for (a) BC, (b) OA, (c) SO$_4^{2-}$, (d) NO$_3^-$ and (e) NH$_4^+$.**



In summary, Figure 17 depicts the median chemical mass composition of $PM_1$ emissions across different ship operational phases, which is the parameter most frequently found to affect EFs. The amount of $PM_1$ emitted by ships can vary by a factor of three depending on the operational phase and tends to be more variable during "manoeuvring/navigation" phases compared to when ships are docked. Particles emitted by ships across all operational phases are primarily composed of organic matter (OA: 75%), black carbon (BC: 21%), and sulphate ($SO_4^{2-}$: 4 %). However, this composition can change with the operational phase: the proportion of black carbon increases to 34% during "manoeuvring", while the proportion of sulphate rises to 8% during "navigation" and decreases to 2% when "at berth".

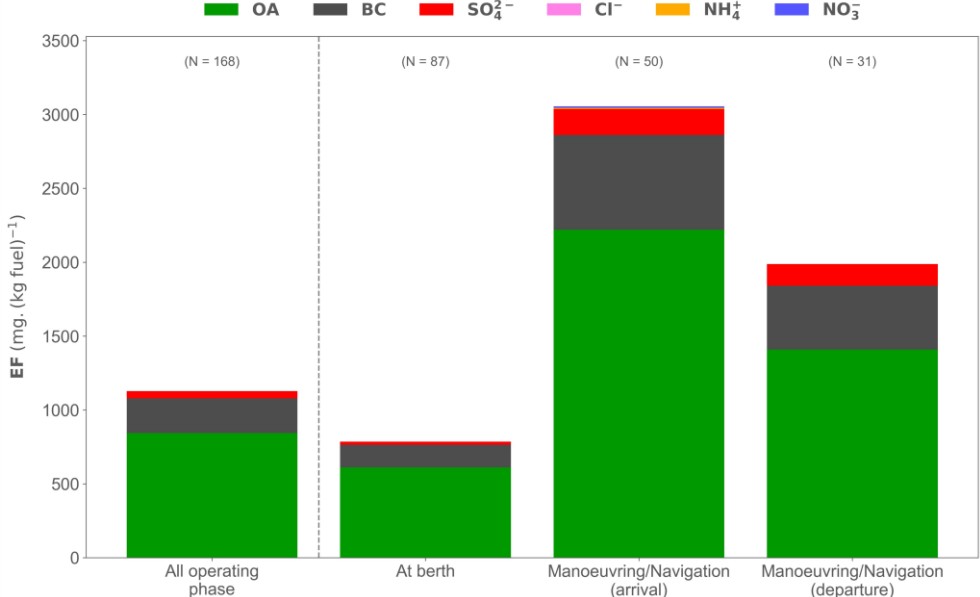

**Figure 17. Median ship $PM_1$ chemical composition as a function of the ship operational phases. The number of plumes considered (N) is specified.**

The $PM_1$ chemical composition characteristics (low sulphate but high organics content) found in the present study share similarities with the "organic-rich" $PM_1$ composition recently identified by PMF analysis of measurements made in Dublin port (Fossum et al., 2024). This suggests that the global ship plumes signature in Marseille port is dominated by ships using VLSFO (Very-Low Sulphur Fuel Oil), ULSFO (Ultra-Low Sulphur Fuel Oil) or MGO (Marine Gas Oil) rather than HFO (Heavy Fuel Oil) fuel combined with a scrubber system, for which sulphate makes up 60% of the $PM_1$ (Fossum et al., 2024). It is noteworthy that in Marseille, the $PM_1$ have a twofold higher BC relative content than the "organic-rich" PM1 detected in Dublin port (21% vs 9%). One reason for its higher value is that both field campaigns experienced different climatic conditions (summertime for Marseille vs wintertime for Dublin). The higher temperature at Marseille (average ambient temperature of 24°C versus 8°C at Dublin) could favour the evaporation of organic from the non-volatile black carbon core of the aerosol once the ship emissions are released in the air.



### 3.2.4. Particle size distribution

To ensure the comparability of the particle size distributions between the different plumes, the emission factors for each class
of particles were normalised with respect to the maximum emission factor observed in each plume. Significant variations in
EFs between plumes require this standardisation.

Among the ship parameters examined, the operational phase has the most significant impact on the particle size distribution
of the 158 selected plumes. For ships "at berth", which operate their auxiliary engines at a stable and optimal load using low-
sulphur distillate fuel (<0.1%), the particle size distribution was found to be unimodal and centred around 30 nm (Figure 18a).
In contrast, vessels "in manoeuvring/navigation", which use their main engines at lower (<25%) and less stable loads, display
a bimodal distribution with modes at 35 nm and 100 nm (Figure 18b). Unlike the number of particles, no significant differences
are observed between arrivals and departures. The 35 nm mode is generally more prevalent, while the intensity of the 100 nm
mode varies among plumes. Notably, the 100 nm mode is particularly pronounced among the ferries, a category of vessels in
the port of Marseille that are known to be partially equipped with scrubbers.

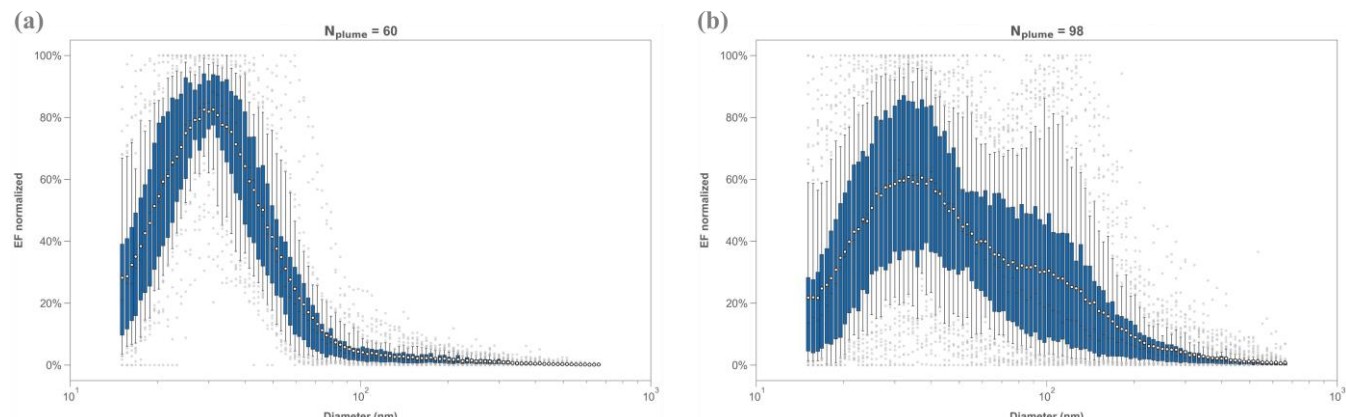


**Figure 18. Particle size distribution of the plumes identified during the campaign according to the operational phase (a) "at berth" and (b) "manoeuvring/navigation".**

These findings are supported by analysis of the daily profiles for mean particle diameter (Figure S6), which shows a shift
towards smaller particles (around 50 nm) during periods of ship movement. These observations are also consistent with the
literature. While the specific causes of a unimodal or bimodal distribution can vary between studies, the presence of the 30 nm
mode is consistently observed. According to particle morphology analyses conducted using transmission electron microscopy
(Aakko-Saksa et al., 2023; Alanen et al., 2020), the 30 nm particles are spherical, non-volatile, and originate from the
combustion of fuel and lubricating oils. The mode around 100 nm could be attributed to (1) incomplete combustion during
manoeuvring phases, which promotes soot formation and particle coagulation (Diesch et al., 2013), (2) the use of scrubbers
when the 100 nm mode is dominant (Jeong et al., 2023; Kuittinen et al., 2021; Winnes et al., 2020), or (3) the use of heavy
fuel oil types (HFO, VLSHFO, ULSHFO) when the 100 nm mode is not dominant (Anderson et al., 2015a; Fossum et al.,
2024). Fossum et al. (2024) demonstrated that (1) plumes from ships "at berth", which used marine gas oil (MGO) fuel, had a





unimodal particle size distribution centred around 30 nm, whereas (2) plumes from ships "in manoeuvring/navigation" within port, which used ultra-low sulphur heavy fuel oil (ULSHFO), exhibited a bimodal distribution consistent with the two modes

observed in the present study. Since the type of fuel and the use of scrubbers were not specified in the AIS database, these hypotheses could not be confirmed.

The analysis of particle modal diameter evolution with plume ageing, as shown in Figure 19, indicates a marginal increase in diameter. This finding suggests that, aside from natural dilution, which gradually reduces particle concentrations within the plumes, no other significant physico-chemical processes occur over the short time scales studied (less than 30 minutes).

However, while the increase in modal diameter is not statistically significant, this observation, along with a statistically significant decrease in $EF_{PN}$, suggests that condensation or coagulation phenomena could occur and contribute to the increase in particle modal diameter.

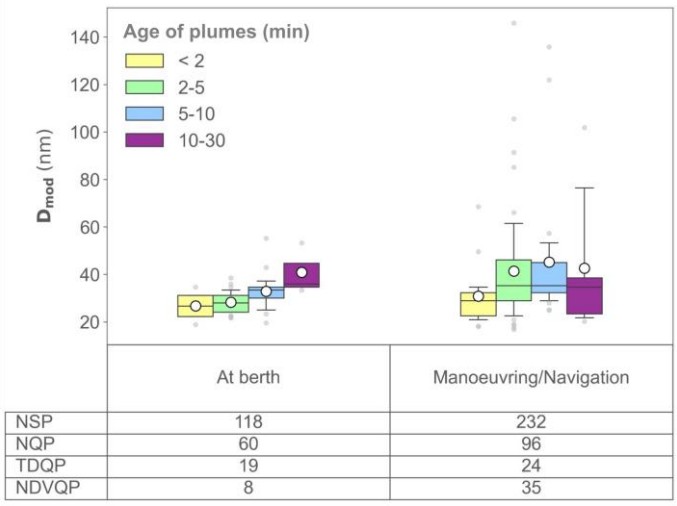

**Figure 19. Distribution of modal particle diameters as a function of ship operational phase and plume age.**


Furthermore, scatter plots of the $EF_{PN}$ measured using the CPC analyser (2.5 nm (PEB) or 7 nm (MAJOR) to 2.5 µm) versus the SMPS analyser (15 nm to 660 nm) yield a slope greater than 1.3 for 30% of the plumes analysed. This finding suggests that, for these plumes, about 30% of the particles had a diameter smaller than 15 nm. Interestingly, the percentage of plumes associated with this feature increases as plume age decreases, consistent with expected plume evolution in the atmosphere

(e.g., 100% of plumes from vessels at berth with an age of less than 2 minutes exhibit this bias, compared to 40% for plumes aged between 2 and 5 minutes).

The analysis of particle size distribution emitted by ships emphasizes the importance of specifically monitoring the $PM_1$ fraction, particularly particles smaller than 150 nm. This range includes the two likely particle size modes, which could be characteristic of the fuels used by ships.





## 3.3. Short-term impact of shipping plumes on local air quality

Following the method described in section 2.3.3, table 2 lists the median, 25th and 75th percentiles enhanced concentration levels of pollutants observed downwind the 353 ship plumes identified during the campaign. metals are included in this analysis to provide insights into their contributions even though this resolution did not allow to determine EF and may underestimate average concentrations during plume events.

**Table 2. Absolute and relative ambient concentration enhancements downwind the ship plumes. The 25th and 75th percentiles are indicated for each median value and are presented as follows median [25th percentile / 75th percentile].**

| | Measured Quantity | Species | units | $N_{plumes}$[1] | Absolute concentrations enhancement[2] | Relative concentration enhancement[3] |
|---|---|---|---|---|---|---|
| **GAS PHASE** | **Nitrogen Oxides** | $NO_X$ | µg.m$^{-3}$ | 328 | 67.1 [38.1 / 96.0] | 68.3% [51.1% / 80.7%] |
| | | NO | µg.m$^{-3}$ | 329 | 25.2 [13.6 / 42.7] | 84.3% [69.9% / 94.2%] |
| | | $NO_2$ | µg.m$^{-3}$ | 328 | 25.9 [15.8 / 37.0] | 58.9% [37.2% / 73.3%] |
| | **Carbon Oxides** | $CO_2$ | ppm | 353 | 3.1 [2.2 / 4.4] | 0.7% [0.5% / 1.1%] |
| | | CO | ppb | 353 | 9.4 [5.6 / 17.0] | 4.1% [<0.01% / 10.8%] |
| | **Sulphur Dioxide** | $SO_2$ | µg.m$^{-3}$ | 286 | 0.6 [<DL[4] / 1.1] | 21.5% [- / 41.3%] |
| | **Ozone** | $O_3$ | µg.m$^{-3}$ | 279 | -20.8 [-12.6 / -31.2] | -48.9% [-26.5% / -82.2%] |
| | **Volatils Organics Compounds (VOC)** | $CH_4$ | ppb | 353 | 1.1 [0.3 / 2.9] | 0.04% [<0.01% / 0.13%] |
| | | $(C_8H_{10})H^+$ | ppb | 132 | 0.03 [0.01 / 0.10] | 0.01% [<0.01% / 18.7%] |
| **PARTICULATE PHASE** | **Particle number** | PN | part.cm$^{-3}$ | 335 | 12 727 [7 204 / 19 338] | 52.1% [39.2% / 63.5%] |
| | **Particle mass concentration** | $PM_{1\ (SMPS)}$ | µg.m$^{-3}$ | 236 | 1.8 [1.1 / 3.3] | 19.4% [<0.01% / 32.5%] |
| | **Chemical Composition (PM$_1$)** | BC | µg.m$^{-3}$ | 342 | 0.47 [0.26 / 0.88] | 42.7% [26.0% / 62.8%] |
| | | $NH_4^+$ | µg.m$^{-3}$ | 178 | 0.02 [0.01 / 0.04] | <0.01% [<0.01% / 4.4%] |
| | | $NO_3^-$ | µg.m$^{-3}$ | 178 | 0.03 [0.01 / 0.04] | <0.01% [<0.01% / 12.8%] |
| | | OA | µg.m$^{-3}$ | 178 | 1.28 [0.74 / 2.92] | 19.7% [12.3% / 34.8%] |
| | | $SO_4^{2-}$ | µg.m$^{-3}$ | 178 | 0.08 [0.04 / 0.25] | 4.3% [1.0% / 12.1%] |
| | **Metals Composition (PM$_1$)** | Ca | ng.m$^{-3}$ | 70 | <DL [<DL / 1.4] | - [- / 4.5%] |
| | | Fe | ng.m$^{-3}$ | 70 | <DL [<DL / 3.4] | - [- / 18.0%] |
| | | K | ng.m$^{-3}$ | 70 | <DL [<DL / 2.0] | - [- / 11.4%] |
| | | Ni | ng.m$^{-3}$ | 70 | 0.9 [<DL / 4.3] | 32.2% [- / 90.7%] |
| | | V | ng.m$^{-3}$ | 70 | 0.5 [<DL / 2.5] | 28.2% [- / 85.1%] |
| | | Zn | ng.m$^{-3}$ | 70 | <DL [<DL / 0.2] | - [- / 4.6%] |

(1) $N_{plumes}$ represents the total number of plumes used as the basis for the statistical calculations; (2) statistics from the average excess concentration of each plume; (3) statistics from the relative contribution of each plume, relative to global concentrations; (4) Below detection limit (<DL).

The level enhancement of pollutants concentration resulting from ship emissions, as detailed in Table 2, represents the supplementary short-term exposure (~10 minutes) that people may be exposed when carrying out activities near the port such as walking, exercising, or dining outdoors. Significant increases in concentration, typically ranging from a factor of 2 to 4, are observed between the 25th and 75th percentiles. These fluctuations are influenced by parameters that play a crucial role in the



dilution of plumes (mainly wind speed and the distance between the measurement station and the ship) in addition to those previously identified as having a significant impact on emission factors (operating phase, ship category, plume age). It is

consequently essential to include many plumes under various meteorological conditions, representative of the area under study, to accurately estimate the impact of ship plume emissions on local air quality.

During plume events, ships significantly contribute to ambient concentrations, with median contributions exceeding 50% for $NO_X$ and PN, greater about 20% for $SO_2$, $PM_1$, BC, OA, Ni, and V and about 4% for CO and $SO_4^{2-}$. Unlike other pollutants, $O_3$ decreases during plume events by an amount nearly equal to the $NO_2$ increase, which results from the reaction between NO

and $O_3$; this decrease contribute to a lowering of the $O_3$ levels by 50%. While the other species exhibit limited median contributions from shipping, some, like aromatic C8 NMVOCs, $NO_3^-$, Fe, and K, occasionally reach significant levels (75th percentiles ranging from 10 to 20%). Meanwhile, concentrations of $CO_2$ and $CH_4$ that are negligible in terms of air quality impact yet continue to be relevant as contributors to greenhouse gases. Enhanced concentrations found in this study for PM, PN, $NO_2$ are 8-15 times higher than those reported by Ausmeel et al. (2020) at a coastal site on the Falsterbo peninsula in

southern Sweden during summer, despite using the same methodology. This discrepancy is partly due to the greater distance between the shipping lanes and the measurement station in the Ausmeel et al. (2020) study (10 km versus 250 m) that further dilutes the plumes. The results of this study are comparable to those reported by Toscano et al. (2022) and Ledoux et al. (2018) from a monitoring stations located in the port of Naples (Italy) and Calais (France), respectively, and located 200 m and 500 m from the shipping lanes. They compared concentrations measured downwind of ship emissions with those from other sectors,

benefiting from well-defined wind sectors. Toscano et al. (2022) reported increases in concentrations of pollutants attributed to passenger ships at hotelling, with NO, $NO_2$, $NO_X$, and $SO_2$ levels rising by 23.4, 23.6, 59.6, and 1.3 µg.m$^{-3}$ respectively. Similarly, Ledoux et al. (2018) observed increases in concentrations of pollutants linked to passenger ships during hotelling and manoeuvring, with NO, $NO_2$ and $SO_2$ levels increasing by 28.4, 28.4, and 16.1 µg/m³, respectively. Regarding $SO_2$, the additional concentrations reported by Ledoux et al. (2018) were significantly higher because their study was conducted in

2014, when sulphur content in fuels (FSC) in this area was still limited to 1%. In contrast, this study and that of Toscano et al. (2022) took place in 2021, when FSC was restricted to 0.1% within port limits and 0.5% outside. This drastic reduction aligns with observations following the shift to fuels with less than 0.5% sulphur in Marseille in 2020 (Figure 4)

Among the trace metals, only vanadium (V) and nickel (Ni) show non-zero median contributions (30%). This corresponds to median additional concentrations of 0.5 ng.m$^{-3}$ and 0.9 ng.m$^{-3}$, respectively, leading to a V/Ni ratio of about 0.5. This ratio

aligns with the ratio that has recently arisen from the use of lower sulphur content fuels since 2020 in all ocean areas. Before 2020, the V/Ni ratio often associated with ship emissions ranged between 2 and 3 (Pandolfi et al., 2011; Viana et al., 2009; Yu et al., 2021), but a shift towards a ratio <2 has been observed. Fossum et al. (2024) observed a V/Ni ratio between 0 and 2, and Yu et al. (2021) reported a ratio of 0.5 (the V/Ni ratios in this study range between 0.1 and 2). Moreover, the analysis of additional metal concentrations categorized by operating phase (Table S13), the parameter identified as having the greatest

influence on emission factors, highlights significant differences depending on the operating phase. For ships "at berth", the additional metal concentrations are systematically non-detected, with only potassium (K) being occasionally detected (75th





percentile = 1.5 ng.m$^{-3}$). For the "in manoeuvring/navigation" phases, excluding ships at berth doubles the median contribution of V and Ni, bringing it to approximately 70%. In addition, as with emission factors, a distinction can also be made between arrivals and departures. A factor of 2 is observed between the additional concentrations during arrivals and departures.

However, this result should be interpreted with caution, as, unlike emission factors, excess concentrations depend on plume dilution, particularly since ship arrivals mainly occur early in the morning when meteorological conditions are less favourable for pollutant dispersion. Separating by operational phase also makes the presence of iron (Fe) in ship emissions during "in manoeuvring/navigation" phases more systematic. For these phases, the median contributions of ships are 11% during arrivals and 5% during departures, compared to 0% across all operational phases.

Our findings highlight the need to incorporate measurements of ultrafine particles (UFP) and their chemical components, including black carbon (BC) and metals, into air quality assessments. These particulate fraction, significantly emitted by ships, are crucial for evaluating human exposure risks (WHO, 2021). Furthermore, quantifying the composition of these particles— particularly BC and metals, which are also released in notable quantities by ships—is essential for assessing the associated health risks to populations (Briffa et al., 2020, p.4; Rönkkö et al., 2023).

**4.  Conclusions**

Emissions from ships significantly play an important role in the exposure of human populations to atmospheric pollutants in port areas. However, these emissions are not well understood, especially since the advancements in ship engine design and the implementation of purification technologies due to regulatory restrictions. Considering these elements, measurement campaign has been conducted in Marseille, one of the largest ports in the Mediterranean Sea, which will become an Emission Control

Area (ECA) for SO$_x$ in 2025. Measurements were taken at two stations within the port area in June 2021, capturing high-resolution data on the chemical composition of both the gaseous phase (e.g., SO$_2$, CO$_2$, NO$_X$, CH$_4$) and particulate phase (e.g., BC, OA, SO$_4^{2-}$), as well as the particle size distribution. In total, nearly 110 compounds were measured simultaneously, including 45 trace metal elements and 41 NMVOCs, creating a unique and comprehensive database.

The comparison of concentrations measured at sites within the port area (PEB and MAJOR) with those from the MRS-LCP

station, which serves as a reference for urban background pollution for the regional air quality network, as well as the analysis of their temporal evolution, clearly demonstrates the impact of port activities on pollutant levels. For particle number (PN), PM$_{2.5}$, PM$_1$, and NO$_X$, the influence is significant, with average concentrations being 1.5 to 2 times higher near the port. In contrast, for pollutants such as SO$_2$ and certain metals (As, Cd, Co, Fe, Ni, Sb, Se, Sn, V, Zn, and Zr), only the peak concentrations are affected, with maxima 2 to 10 times higher near the port compared to downtown areas.

In addition, a method based on cross-referencing measurement data with meteorological and AIS records, which include ship positions, was developed and applied to identify ship emission plumes and characterize their physical and chemical composition. In total, more than 350 ship plumes were identified to determine the emission factors (EFs) of particle- and gas-phase species. These EFs were calculated as quantities that account for plume dilution and refer to the amount of burned fuel.



Generally, the ship EFs determined in this study, which primarily cover their port operations (docking, berthing, and entering
or leaving the port), are consistent with the values documented in the literature.

These EFs were also used to explore how various ship-related characteristics influence emissions. The study found that the
operational phase of the ships is the most influential parameter, related to the fact that ships "at berth" operate their auxiliary
engines at a stable and optimal load using low-sulphur distillate fuel (<0.1%), while those "n manoeuvring/navigation" use
their main engines at lower (<25%) and less stable loads. Additionally, the ship categories and the age of the plume can also
affect these EFs.

In terms of chemical composition, ship gaseous emissions are predominantly composed of $NO_X$ (86%) and CO (12%). $SO_2$
and $CH_4$ each represent about 1%. Other compounds, such as NMVOCs, constitute less than 0.1% of the gaseous phase but
can account for up to 10% under certain operational conditions, noting that the impact of these species on secondary pollution
can be significant. Regarding the particulate phase, the quantity of particles emitted by ships can vary by a factor of 3 depending
on the operational phase, with higher emissions during "manoeuvring/navigation" phases compared to "at berth". Particles
emitted by ships, across all operating phases, are mainly composed of organic aerosol (OA) (75%), black carbon (BC) (21%),
and sulphate ($SO_4^{2-}$) (4%). The proportion of BC and sulphate increases to 34% and 8%, respectively during "manoeuvring"
and "navigation" phases. The study also shows that combining $PM_1$ size distribution analysis with their chemical composition
(organic fraction, sulphate, black carbon, and vanadium/nickel ratio) can help identify the different types of fuels used by ships
as well as the exhaust gas cleaning systems installed on vessels.

Finally, the additional concentrations from ship emissions represent the supplementary short-term exposure (~10 minutes) that
people may experience when carrying out activities near the port. This indicates that during a plume event, ships contribute
significantly to ambient concentrations of certain pollutants, with median contributions exceeding 50% for $NO_X$ and particle
number (PN), around 20% for $SO_2$, $PM_1$, black carbon (BC), organic aerosols (OA), nickel (Ni), and vanadium (V), and
approximately 4% for CO and $SO_4^{2-}$. A more detailed analysis conducted on metals, for which the estimation of EFs was not
possible due to the analyser's resolution time, highlights that metals are not appropriate tracers of ship pollution when ships
are at berth. However, certain metals such as V, Ni, and Fe appear to be good tracers during the "manoeuvring/navigation"
phases. The median V/Ni ratio of 0.5 obtained is consistent with the new ratios established following the use of lower sulphur
content fuels since 2020 in all oceans.

The results from this study provide robust support for assessing air quality in port areas, improving emission inventories—
particularly in terms of chemical speciation—and source apportionment. They can also serve as a baseline for studying the
benefits of implementing an Emission Control Area (ECA) for $SO_X$ in the Mediterranean Sea in 2025.



*Supplementary data.* The supplement related to this article is available online at: awaiting DOI.

*Data availability.* Data sets concentrations from the study are available at https://doi.org/10.57932/90ffebbe-94c3-4356-a073-78ec9e014b1d (Le Berre et al., 2024). Toolkits used in this study are available upon request from the authors Brice Temime-Roussel (brice.temime-roussel@univ-amu.fr) or Lise Le Berre (lise.le-berre@univ-amu.fr).


*Author contributions.* NM, SS and HW designed the measurement campaign. BD'A took responsibility for the scientific coordination of the field campaign, with AA, BTR, LLB and GML contributing to the search for measurement stations and obtaining authorisation to install the measurement instruments in the port. The measurements and their processing conducted by BTR, LLB, GML, SS, LT, TL, JRB, GG, LL and RB. LLB carried out data analysis (compilation of databases, development
of the peak detection algorithm, development of a toolkit for estimating emission factors, analysis of plume composition and parameters influencing it) and wrote the manuscript. BTR and HW reviewed the paper. All authors have read and approved the submitted version of the manuscript.

*Declaration of competing interest.* The authors declare that they have no known competing financial interests or personal
relationships that could have appeared to influence the work reported in this paper.

*Acknowledgments.* The authors would like to warmly thank the Port of Marseille for allowing us to install the measuring instruments on the berth throughout the campaign. In particular, we are grateful to Eric Beroule, Mickaël Parra and Magali Deveze for their administrative and technical support in preparing and carrying out the PAREA field campaign. The authors
gratefully acknowledge the MASSALYA instrumental platform (Aix Marseille Université, lce.univ-amu.fr) and MRS-LCP background urban supersite of Marseille (AtmoSud) for the provision of measurements used in this publication. They would also like to thank Irène Xueref-Rémy from IMBE laboratory for the loan of her $CO/CO_2$ analyser (PICARRO G2401) and standard gas cylinders, and for her technical support with it.

*Financial support.* The measurement campaign was supported by the French Agency for Ecological Transition (ADEME) as part of the PAREA project of the CORTEA research programme (grant no. 1966C0015). The data analysis received financial support from the Provence-Alpes-Côte-d'Azur regional air quality monitoring network (AtmoSud) and ADEME through the funding of LLB's PhD (grant no. TEZ19-029). The exploratory campaigns to determine the station locations were supported by European Union's Horizon 2020 research and innovation programme under grant agreement No. 814893 (the project
"Shipping Contributions to Inland Pollution Push for the Enforcement of Regulations", SCIPPER).



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
