# Peer review of "Measurement report: In-depth characterization of ship emissions during operations in a Mediterranean port"

_EGUsphere, 2024_

## Author Response (AR2)

**RC1**

We sincerely thank the referee for taking the time to evaluate our manuscript and for providing positive and constructive feedback, which has significantly contributed to improving the quality of the paper. Our detailed responses to the comments are presented below.

▶ **Major comments**

• *Line 363 mentions using a statistical test to determine differences between different ship/operation categories, however, the rest of the manuscript only occasionally mentions statistical differences. Many of the figures show results that look similar, and it would be useful to identify which differences are statistically significant.*

**Response:**

To address this comment, we have carefully revised the manuscript to ensure that every comparison explicitly states whether the difference is statistically significant. These clarifications have been incorporated throughout the revised manuscript (lines 419–421, 431, 435, 437, 442, 455, 457, 479, 496, etc.).

• *There are a lot of figures, and some of them show similar data (i.e figure 11 shows the same data as figure 12 and 13, just not broken down by different categories. Same with figure 14, 15 and 16). The authors might consider if there is a way to combine some of these figures.*

**Response:**

Figures 11 and 14 provide an overall view of emission factor variability across all vessels, allowing direct comparison with literature data. In contrast, Figures 12–13 and 15–16 provide a more detailed analysis based on operational phase, ship category, and plume age.

Combining Figures 12–13 and 15–16 as suggested by the reviewer, was initially considered during the manuscript preparation. However, doing so would require categorizing the data into 48 sub-classes, given the number of plumes analysed (~350), which is already a substantial dataset for this type of study. Even under an ideal scenario of equal distribution, this would result in approximately 7 plumes per category, which remains statistically weak. Consequently, despite the relatively large number of plumes analysed, the statistical analysis of variations between categories would lack robustness and could not be reliably performed.

For these reasons, we believe that maintaining the current figure structure provides a clearer and more statistically meaningful representation of the results.

▶ **Minor comments**

• *Line 58: Although there are some details on the MARPOL convention regulations, a few additional clarifications would be useful to expand these results to other locations. For example.*
*- Is the port in an ECA, and how far offshore does this extend.*
*- Please expand on the usage of SCR systems. It is discussed that these allow for higher sulphur fuels in the ECA, are these also required offshore? Is there any information on how common these systems are? It is also noted that these scrubbers are shut down in the harbor (line 445) - Does*

*this increase the sulphur emissions?*

**Response:**

- Figure A1 in the reply supplement illustrates the locations of existing ECAs in 2021 (campaign period). The Mediterranean Sea, including Marseille, will only become an Emission Control Area (ECA) for $SO_X$ from May 2025 under MARPOL Annex VI Regulation 14. All ships operating in the Mediterranean Sea must either use fuels with a sulphur content below 0.1% or adopt emission control technologies that achieve an equivalent emissions level. However, before this date and since 2010, ships at berth or at anchor for more than two hours were subject to European regulations, that imposed the same emission limits as those in ECAs (<0.1%). At sea ships in the Mediterranean had to comply with the global IMO sulphur cap of 0.5% that has been effective since 2020. To enhance clarity and highlight the relevance of our results to both ECA zones when ships are docked and other areas during navigation, we have included the following paragraph at line 123:

  "In 2021, the port of Marseille was not within an Emission Control Area (ECA) and followed the global sulphur cap of <0.5%. However, EU regulations required ships docked for less than two hours to limit sulphur emissions to <0.1%, as in ECAs. From May 2025, the Mediterranean, including Marseille, will become an ECA (ECAMED), enforcing a <0.1% sulphur limit for all vessels operating within it."

- SCR systems and scrubbers serve distinct functions and regulate different types of emissions. SCR systems are specifically designed to reduce nitrogen oxide (NOx) emissions by converting NOx into nitrogen and water through a catalytic reaction using ammonia or urea as a reducing agent. They do not impact sulphur emissions and do not allow the use of higher-sulphur fuels in ECAs. In contrast, scrubbers (exhaust gas cleaning systems) are designed to reduce sulphur oxide ($SO_X$) emissions. The use of scrubber enables ships to continue to operate on high-sulphur fuels while complying with SOx regulations, both at sea and in port. The selection of the emission control technologies, including SCR systems and scrubbers, varies significantly depending on vessel age, category, and operational region. Newer vessels, particularly those frequently operating in ECAs, are more likely to be equipped with SCR systems due to stricter NOx compliance requirements. By 2017, nearly 2,000 ships had installed SCR systems (Zannis et al., 2022). As of 2020, approximately 4,300 ships worldwide were equipped with scrubbers (Hassellöv, 2020), representing about 4% of the global fleet. However, vessels operating in Marseille, particularly ferries, are known to be well-equipped with scrubbers. It is important to note that information on onboard emission control systems remains difficult to obtain and is not included in the Automatic Identification System (AIS) vessel tracking database.

  Regarding scrubber deactivation in port, this hypothesis could explain why $SO_2$ emission factors tend to be higher upon vessel arrival compared to departure. Indeed, as highlighted by Teinilä et al. (2018), a short but intense emission peak is observed upon port arrival when the scrubber is switched off. However, this switch-off primarily applies to open-loop scrubbers, which discharge treated wash water directly into the sea. Additionally, scrubber deactivation is often accompanied by a fuel switch, transitioning from high-sulphur heavy fuel oil (HFO) with a scrubber to low-sulphur marine gas oil (MGO). This peak is attributed to the purging of the

engine system during the transition process.

To address these points in the manuscript, we have made the following modifications:

- o **Clarified the distinction between SCR systems and scrubbers in the main text** at line 68-70 by amending sentences as follow :

  "These regulations have led to significant progresses in ship engines and to the introduction of after-treatment devices, such as Selective Non-Catalytic Reduction (SCR) systems and scrubbers, which serve distinct functions. SCR systems reduce $NO_X$ emissions through a catalytic reaction and do not impact sulphur emissions. In contrast, scrubbers remove sulphur from exhaust gases, allowing the use of high-sulphur fuels exceeding 0.5% or 0.1% while complying with $SO_X$ regulations. Open-loop scrubbers discharge treated wash water into the sea, whereas closed-loop systems recycle it. These technologies are sometimes combined to meet both $NO_X$ and $SO_X$ regulations in ECAs."

- o **Provided additional information regarding scrubber deactivation in port** at line 446 as follow :

  "When this type of scrubber is shut down, a fuel switch from HFO to MGO is typically performed beforehand. However, a temporary increase in $SO_2$ emissions can be observed (Teinilä et al., 2018), attributed to engine system purging during the transition process."

- *Line 292: "As shown Figure 3, CPBF indicates that the highest concentrations typically occur when the measurement sites are downwind of the mooring berths or the ships' access lanes to the port". Where are the referred to berths and access lanes. Is this to the northwest of the CPF site?*

**Response:**

The locations of the port quays and access channels are illustrated in Figure S2 in the preprint, along with the measurement station positions. The PEB station is influenced by the main access channel, which follows a northwest-southeast axis. In contrast, the MAJOR station is influenced by an access channel that curves, following both northwest and southwest directions. The relative positions of the quays also vary depending on the measurement station:

- At PEB station, cruise terminals are located across a broad northwest sector, the container terminal is to the north, and the ferry berths for Corsica are to the southeast.

- At MAJOR station, the cruise terminal is to the southwest, the ferry berths for Corsica are to the west, and the quays serving vessels to and from North Africa are to the northwest.

**To reflect this, a reference to Figure S2** has been added in the revised manuscript at line 292: "(located in Figure S2)"

◗ **Typographical comments**

- *Line 79: The acronym MGO is not defined*

  **Response:**

  The acronym has been specified on line 79 as follows:

  "marine gas oil (MGO)".

[Figure]

**Figure A1.** Map of current Emission Control Areas (ECAs) (filled orange area), including The North American area, The Baltic Sea area, the Baltic Sea, The North Sea area and The Unites States Caribbean (IMO, 2021).

**RC2**

We greatly appreciate the referee's time and effort in reviewing our manuscript. Their insightful and constructive feedback has been invaluable in enhancing the quality of the paper. Below, we provide detailed responses to the comments.

▶ **Major comments**

- *Line 217: The atmospheric background was calculated using a low pass time series. Then for tuning, plumes without returning to the baseline were not considered in the analysis. How is the baseline considered and what is the impact of the background calculation on it?*

  **Response:**

  The background mentioned between lines 197 and 207 refers to a general background level, used solely for the identification and selection of ship-related peaks. This low-pass filter describes the variability in background concentration caused by atmospheric physico-chemical processes and regional pollutant transport while excluding short-term fluctuations associated with passing ships. In contrast, the baseline return mentioned at line 217 is defined locally around each peak, based on the 30 seconds of measurements before and after the peak, as specified in lines 238–241. Therefore, the background calculated using the low-pass filter has no influence on the determination of this local baseline.

  To clarify this distinction, line 217 has been revised as follows:

  "Plumes that could not be individualized by a return to baseline level in the quantification phase of plume characteristics, as explained in Section 2.3.2, were removed."

- *Line 305: For OA, there is an increase in the average concentrations during the night, which the authors associated to the influence of the land breeze regime. Can that be a consequence of the nighttime radical chemistry? Then, for the NMVOCS, concentrations also increased at night, if those are coming from urban emissions, can the authors comment about the behavior of the other anthropogenic pollutants measured?*

  **Response:**

  We appreciate this insightful comment. In the manuscript, we attribute the initial increase in OA concentrations to the land breeze effect, which transports urban and industrial pollutants toward the port area. This interpretation is supported by the timing of the observed increases, which start around 19:00, aligning with the transition from sea breeze to land breeze. This change in wind direction facilitates the transport of accumulated urban emissions towards the measurement site. Since darkness sets in around 21:00 during this season, the role of nighttime $NO_3\cdot$ chemistry is likely limited during the early evening hours when the increases are observed. The ongoing rise in pollutant levels after 21:00 is likely driven by a lowered boundary layer height, rather than active oxidation processes. The simultaneous rise in BC, and toluene in parallel to OA during this period further supports the dominance of pollutant transport and accumulation.

*Line 328 and 450: What is the impact of the criteria of exclusion of plumes into the analysis of the selected ones? Are they representative of traffic periods?*

**Response:**

We appreciate this question, as it is important to assess the impact of exclusion criteria on the analysis and the representativeness of the selected plumes.

The exclusion criteria have inevitably reduced the number of available plumes, but they ensure a more robust dataset by minimizing uncertainties. The primary goal of this study was to determine emission factors (EFs) representative of the different ship categories operating in the port of Marseille, while avoiding potential biases related to misattribution of plumes to multiple vessels. What matters is that we successfully captured all major ship categories accessing the port. The representativity of overall traffic is not critical in the context of EF determination, as the objective is not to quantify the contribution of each vessel type but rather to establish robust emission factors per category. However, traffic representativity would indeed be a key consideration in studies focusing on environmental concentration impacts, which was not the purpose of this work.

The only notable limitation is the underrepresentation of cruise ships in navigation, due to COVID-19 restrictions during the measurement period. This affects the robustness of EFs for cruise ships in navigation but, conversely, improves the robustness of EFs for cruise ships at berth, as the exceptionally high number of stationary cruise ships during this period provided a rare opportunity to refine EF estimates for this category.

**To clarify this, we have added sentence in line 329:**

"All these precautions ensure the robustness of the emission factors (EFs), especially when analysing EFs as a function of ship characteristics. These restrictions do not affect the EF values themselves, as the number of detected plumes does not influence EF values, which are defined through normalization by $CO_2$."

- *Line 835: Authors mentioned this study provides robust support for assessing air quality in the ports and along the text, they highlight the influence of the sulfur content on the emissions factors. Therefore, can the authors conclude on the limitations of the plume selections and the impact of the use of different fuels and ship classifications on this observation?*

**Response:**

As stated in our previous response, the selection of plumes does not impact the emission factor (EF) values themselves. Regarding the influence of sulphur content, our study does not consider it as the sole determinant of EFs. Instead, throughout the manuscript (lines 459, 538, 590, 619, 664, 687), we emphasize that vessel category (liked with engine power, ship size, and technology) and the ship's operational phase plays a crucial role, encompassing not only fuel sulphur content but also engine type (main vs. auxiliary), engine load conditions (low/unstable vs. optimal/stable), and vessel category (liked with engine power, ship size, and technology). Since the exact fuel type used by each vessel was not available, our analysis is based on measured sulphur emissions rather than fuel specifications. This strengthens the relevance of our findings, as they are grounded in real-world emission profiles rather than declared fuel compositions. To clarify this, we propose modifying the sentence in line 835 as follows:

"The results from this study provide robust support for assessing air quality in port areas and improving source apportionment through enhanced emission profiles. The determined EFs allow for the integration of chemical speciation into emission inventories based on ship operational phases and categories, enabling a more precise estimation beyond traditional approaches relying mainly on engine power or fuel consumption."

▶ **Technical details**

- *Line 183 : Data records were purchased from (incomplete)*
  **Response:**
  The reference was incorrectly formatted as a source rather than as a citation. The sentence has been corrected in the manuscript at line 183 as follows:
  "... data records were purchased from MarineTraffic (2020)."

- *Line 678: PM1, add subindex*
  **Response:**
  Error corrected in revised manuscript by : "$PM_1$"

- *Check acronyms definitions as some are defined multiple times: OA, BC*
  **Response:**
  Thank you for pointing this out. We have carefully reviewed the manuscript and corrected the redundant definitions, including OA and BC, ensuring that each acronym is defined only once at its first occurrence in the text, excluding the abstract.

- *Line 813: check spelling for "n manouevring"*
  **Response:**
  Error corrected in revised manuscript by : "manoeuvring/navigation"

- *Table S11 and Table S6 are hard to read, can the authors increase the size?*
  **Response:**
  Size increased in revised manuscript.